



# Using multiple methods to understand groundwater recharge in a semi-arid area

Shovon Barua[1], Ian Cartwright[1], P. Evan Dresel[2], Edoardo Daly[3]

[1]School of Earth, Atmosphere and Environment, Monash University, Clayton, Victoria 3800, Australia

[2]Agriculture Victoria, Department of Jobs, Precincts and Regions, Bendigo, Victoria 3554, Australia

[3]Department of Civil Engineering, Monash University, Clayton, Victoria 3800, Australia

*Correspondence to*: Shovon Barua (shovon.barua@monash.edu)





# Abstract

Understanding recharge in semi-arid areas is important for the sustainable management of groundwater resources. This study focuses on estimating groundwater recharge rates and understanding the impacts of land-use changes on recharge in a semi-arid area. Two adjacent catchments in southeast Australia were cleared ~180 years ago following European settlement; in one of these catchments eucalypt plantation forest was subsequently established ~20 years ago. Chloride mass balance yields recharge rates of 0.2 to 61.6 mm yr$^{-1}$ (typically up to 11.2 mm yr$^{-1}$). The lower of these values probably represent recharge rates prior to land clearing, whereas the higher likely reflects recharge rates following initial land clearing. The low pre-land clearing recharge rates are consistent with the presence of groundwater that has residence times that are up to 24,700 years (calculated using radiocarbon) and the moderate to low hydraulic conductivities (0.31 to 0.002 m day$^{-1}$) of the clay-rich aquifers. Recharge rates estimated from tritium activities and water table fluctuations reflect those following the initial land clearing. However, recharge rates estimated using water table fluctuations (15 to 500 mm yr$^{-1}$) are significantly higher than those estimated using tritium renewal rates (0.01 to 89 mm yr$^{-1}$; typically <14.0 mm yr$^{-1}$). The higher recharge rates from the water table fluctuations approach the long-term average annual rainfall (~640 mm yr$^{-1}$) and are unlikely to be correct given the estimated evapotranspiration rates of 500 to 600 mm yr$^{-1}$. While it is difficult to examine the uncertainties associated with the water table fluctuation method, it is likely that a reduction in the effective specific yield due to the presence of moisture in fine-grained soils results in the water table fluctuation overestimating recharge rates. Although land-use changes increased recharge rates, the preservation of old groundwater indicates that present-day recharge is generally modest, which is likely to be the case in semi-arid regions of southeast Australia.





# 1. Introduction

Groundwater is a critical resource for meeting the expanding urban, industrial and agricultural water requirements, especially in semi-arid areas that lack abundant surface water resources (de Vries and Simmers, 2002; Siebert et al., 2010). Determining recharge rates is vital for understanding regional hydrogeology and assessing the sustainability of groundwater resources.

Recharge is the water that infiltrates through the unsaturated zone to the water table and thus increases the volume of water stored in the saturated zone (Lerner et al., 1990; Healy and Cook, 2002; Scanlon et al., 2002). A distinction between gross and net recharge may be made (Crosbie et al., 2005). The total amount of water that reaches the water table is the gross recharge, while net recharge accounts for the subsequent removal of water from the saturated zone by

evapotranspiration. In areas with shallow water tables and deep-rooted vegetation, this subsequent water loss can be considerable.

Land-use changes may modify recharge rates and affect groundwater resources (Foley et al., 2005; Lerner and Harris, 2009; Owuor et al., 2016). In many semi-arid regions, there has been the conversion of native forests to agricultural land (Foley et al., 2005). Deep-rooted trees

generally return more water to the atmosphere via transpiration than shallow-rooted crops and grasses (Hewlett and Hibbert, 1967; Bosch and Hewlett, 1982; Fohrer et al., 2001). In southeast Australia, the reduction in evapotranspiration following land clearing has commonly resulted in a net increase in recharge and a rise of the regional water tables (Allison et al., 1990). Eucalyptus tree plantations were subsequently initiated partially to reduce groundwater

recharge and thus reduce waterlogging and salinization of cleared land in southeast Australia by reversing the rise of the regional water tables (Gee et al., 1992; Benyon et al., 2006).

## 1.1 Quantifying groundwater recharge

Estimating groundwater recharge rates is not straightforward (Lerner et al., 1990) and recharge rates in semi-arid areas potentially vary in space and time (Sibanda et al., 2009). There are



many techniques used to estimate groundwater recharge, including measurement of water

infiltration using lysimeters installed in the unsaturated zone, calculating catchment water

budgets, use of remote sensing, application of numerical models, measuring water table

fluctuations, chemical (Cl) mass balance calculations, and/or from the concentrations of

radioisotopes such as $^3$H (tritium), $^{14}$C (carbon), $^{36}$Cl (chloride) or other time-sensitive tracers

in groundwater, e.g., chlorofluorocarbons (Scanlon et al., 2002, 2006; Healy, 2010; Doble and

Crosbie, 2017; Cartwright et al., 2017; Gelsinari et al., 2020). Understanding which methods

are suitable to determine groundwater recharge is critical to understand the effects of successive

land-use changes on groundwater recharge.

Different techniques estimate recharge over different spatial-temporal scales, and they may

thus yield different results (Scanlon et al., 2002). In addition, each technique has different

uncertainties. Therefore, studies often recommend utilizing multiple methods to constrain

recharge (Healy and Cook, 2002; Sophocleous, 2004; Scanlon et al., 2006). Understanding the

broader hydrogeology also helps to understand recharge. For example, catchments where

recharge rates are high should contain high proportions of young groundwater. Additionally,

recharge rates are likely to be low if evapotranspiration rates approach rainfall totals. A brief

description of the techniques used in this study is provided highlighting assumptions and

limitations of each method.

### 1.1.1 Cl mass balance

The Cl mass balance (CMB) approach yields average regional net recharge rates (Bazuhair and

Wood, 1996; Scanlon, 2000). The assumptions of this method are that all Cl in groundwater

originates from rainfall and that Cl exported in surface runoff is negligible or well known.

Under these conditions, the net groundwater recharge ($R_{net}$ in mm yr$^{-1}$) is estimated from:

$$R_{net} = P \ \frac{Cl_p}{Cl_{gw}} \tag{1}$$





(Eriksson and Khunakasem, 1969) where $P$ is mean annual precipitation (mm yr$^{-1}$), $Cl_p$ is the

weighted mean Cl concentration in precipitation (mg L$^{-1}$), and $Cl_{gw}$ is Cl concentration in

groundwater (mg L$^{-1}$). The CMB method estimates net recharge rates averaged over the time

period over which the Cl contained within the groundwater is delivered; this may be several

years to millennia. Uncertainties in the CMB method are mainly the uncertainties in the long-

term rate of Cl delivery and the assumptions that runoff has remained negligible over time.

**1.1.2 Water table fluctuations**

Water table fluctuations may be used to estimate gross recharge over the time period for which

groundwater elevation data is available. The water table fluctuation (WTF) method strictly

requires the water table to be located within the screened interval of the bore; however, it can

be used in bores screened within a few metres of the water table (Healy and Cook, 2002). The

method assumes that: evapotranspiration from the water table has not occurred; the rise in the

water table is solely due to recharge following rainfall events; groundwater elevations are not

influence by pumping; and the water table falls in the absence of recharge. $R_{gross}$ is calculated

from

$$R_{gross} = S_y \frac{\Delta h}{\Delta t} \qquad (2)$$

where $S_y$ is the specific yield (dimensionless) of the aquifer, and $\Delta h/\Delta t$ is the variation in the

hydraulic head over the recharge event (mm yr$^{-1}$ where there is an annual recharge event).

Despite its simplicity, there are several potential uncertainties in the WTF method. $S_y$ is not

commonly measured, and most studies rely on typical values based on aquifer materials.

Additionally, $S_y$ is commonly viewed as a constant aquifer property ("the volume of water that

an unconfined aquifer releases from storage per unit surface area of aquifer per unit decline in

the water table": Freeze and Cherry, 1979). However, that ignores the moisture in the

unsaturated zone held in and above the capillary fringe (Gillham, 1984; Crosbie et al., 2005,

2019), which results in $S_y$ varying with depth and over time (Childs, 1960). The presence of



this soil moisture reduces the volume of water needed to saturate the aquifer matrix, thus

reducing the effective $S_y$. If the soil becomes fully saturated due to the rise of the capillary

fringe, it has an effective $S_y$ close to 0, and small recharge events can produce significant and

rapid increases in the head (Gillham, 1984). This can be an issue in areas of fine-grained soils

where the capillary fringe may be several metres thick. Not considering the potential reduction

in the effective $S_y$ in areas of shallow water tables leads to recharge being overestimated by the

WTF method. Other processes may also affect head measurements, such as entrapment of air

during rapid recharge events (the Lisse effect) and the impacts of barometric pressure changes

and ocean or Earth tides, especially when the head is measured using sealed pressure transduces

(Crosbie et al., 2005). The estimation of the recession curve of the groundwater hydrograph

used to calculate $\Delta h$ in Eq. (2) also involves some judgement.

**1.1.3 $^3$H renewal rate**

The $^3$H renewal rate (TRR) method envisages that recharge mixes with pre-existing

groundwater at the top of the aquifer. The renewal rate ($R_n$) represents the proportion of new

water added in each recharge cycle with an equivalent amount displaced lower into the

groundwater system. If there is an annual cycle of groundwater recharge, the $^3$H activity of

groundwater in year $i$ ($^3H_{gw_i}$) is related to $R_n$ by

$$^3H_{gw_i} = (1-R_{n_i})^3H_{gw_{i-1}}e^{\lambda_t} + R_{n_i}{}^3H_{p_i} \tag{3}$$

(Leduc et al., 2000; Le Gal La Salle et al., 2001; Favreau et al., 2002) where $\lambda_t$ is the radioactive

decay constant for $^3$H (0.055764 year$^{-1}$), and $^3H_{p_i}$ is the average $^3$H activity of rainfall in year $i$

(in Tritium Units, TU where 1 TU corresponds to $^3$H/$^1$H = $1\times10^{-18}$). The application of the

TRR method requires the $^3$H input function over the past few decades to be known. The $^3$H

activities of southern hemisphere groundwater recharged during the 1950s and 1960s

atmospheric tests were several orders of magnitude lower than northern hemisphere

groundwater (Morgenstern et al., 2010; Tadros et al., 2014). These $^3$H activities have now





decayed and are lower than those of present-day rainfall, which results in individual $^3$H

activities yielding a single $R_n$ estimate (Cartwright et al., 2007, 2017), which is not yet the case

in the northern hemisphere (Le Gal La Salle et al., 2001).

Groundwater recharge rates are related to $R_n$ by

$$R_{net} = R_n bn \qquad (4)$$

where $b$ is the thickness of the upper part of the aquifer system that receives annual recharge

and $n$ is the effective porosity. Uncertainties in the TRR estimates include uncertainties in the

$^3$H input function and having to estimate $b$ and $n$, which may be variable and not well defined.

The recharge rates are net estimates averaged over the residence time of the groundwater in the

upper part of the aquifer, which is approximately $R_n^{-1}$.

### 1.2 Objectives

This study estimates recharge rates using the Cl mass balance, water table fluctuations, and $^3$H

renewal rate methods in a semi-arid area that has undergone successive land-use changes. We

evaluate the applicability and uncertainties of these commonly applied methods to determine

the changes in recharge rates caused by these successive land-use changes. While based on a

specific area, the results of this study, in particular the comparison of the estimates of recent

recharge rates, will be applicable to similar semi-arid areas elsewhere.

## 2. Study area

Gatum is situated in western Victoria, southeast Australia (Fig. 1a). The native eucalyptus

forests in this region were originally cleared for grazing following European settlement ~180

years ago (Lewis, 1985) and then partially replaced by eucalyptus plantation in the last ~20

years (Adelana et al., 2014). Gatum lies in the regional recharge area of the Glenelg River

Basin to the south of the drainage divide between the Glenelg and Wannon Rivers, and surface

water drains to the Wannon River via the Dundas River (Dresel et al., 2012). The area is

predominantly composed of Early Devonian ignimbrites containing abundant large locally



derived clasts near their base (Cayley and Taylor, 1997). Post-Permian weathering has

produced a deeply weathered saprolitic clay-rich regolith and ferruginous laterite duricrust

(Brouwer and Fitzpatrick, 2002). Some of the drainage areas contain Quaternary alluvium and

colluvium (Adelana et al., 2014).

The study area consists of two catchments with contrasting land-use, one catchment is

predominately dryland pasture used for sheep grazing, and the other is mostly occupied by

plantation *Eucalyptus globulus* forestry. The pasture catchment is around 151 ha and is typical

of the cleared land in this region. It is covered by perennial grasses with about 3 % remnant

eucalyptus trees. The forest catchment is around 338 ha and comprises approximately 62 %

plantation forest, established in 2005, and 38 % grassland (Adelana et al., 2014). The elevations

of the pasture and forest catchments range from 236 to 261 m and 237 to 265 m AHD

(Australian Height Datum), respectively. The two catchments were subdivided into the upper

slope, mid-slope and lower slope, based on the elevation of the study area; the drainage zones

are in the riparian zones of the small streams (Dresel et al., 2018). The catchments are drained

by two small intermittent streams (Banool and McGill Creeks: Fig. 1a) that export ~8 % of

annual rainfall (Adelana et al., 2014; Dresel et al., 2018). In addition to the regional

groundwater system, shallow (1 to 4 m deep) perched groundwater exists in the riparian zones

(Brouwer and Fitzpatrick, 2002; Adelana et al., 2014).

The climate is characterized by cool, wet winters and hot, dry summers. From 1884 to 2018,

the average annual rainfall at Cavendish (Station 089009) ~19 km south of Gatum was ~640

mm (Bureau of Meteorology, 2020), with most rainfall in the austral winter between May and

October. Average annual actual evapotranspiration across the two catchments between 2011

and 2016 was estimated at about 580 mm (Dresel et al., 2018). The mean concentrations of Cl

in rainfall range from 2.2 mg L$^{-1}$ at Cavendish (Hutton and Leslie, 1958) to 4.4 mg L$^{-1}$ at

Hamilton (~34 km south of Gatum: Bormann, 2004; Dean et al., 2014). Similar Cl


concentrations are recorded in rainfall across much of southeast Australia (Blackburn and

McLeod, 1983; Crosbie et al., 2012).

## 3. Methods and Materials

### 3.1 Water sampling

The two catchments at Gatum were instrumented in 2010. There are 19 monitoring bores at

different landscape positions sampling the regional groundwater in the pasture and forest

catchments (Fig. 1a) with sampling depths ranging from 1.3 to 29.7 m (Supplementary Table

S1). Hydraulic heads have been measured since 2010 at four hourly intervals using In Situ

Aquatroll or Campbell CS450 WL pressure loggers corrected for barometric pressure

variations using In Situ Barotroll loggers. Occasional spikes (generally resulting from the

logger being removed from the bores) were removed. Twelve shallow piezometers (~1 m deep

with ~10 cm wide screens at their base) were installed in 2018 near the monitoring bores in the

drainage zones and the lower slopes of the pasture and forest catchments (Fig. 1a). These

piezometers sample the riparian groundwater that in places is perched above the regional

groundwater. Regional groundwater was sampled from the bores (n = 24) and riparian

groundwater from shallow piezometers (n = 24) between May and November 2018. The

groundwater samples were collected from the screened interval using a submersible pump or

bailer following the removal of at least three bore volumes of groundwater or removing all

water and allowing it to recover. Following sampling, hydraulic conductivities ($K_s$: m day$^{-1}$)

were determined from the rate of recovery of the groundwater levels measured at 3-minute

intervals using an In Situ Aquatroll pressure logger (Hvorslev, 1951).

### 3.2 Analytical techniques

Geochemical data are presented in Table S1. Electrical conductivity (EC) was measured in the

field using a calibrated hand-held TPS WP-81 multimeter and probe. Groundwater samples

were collected in high-density polyethylene bottles and stored at ~4°C prior to analysis.





Alkalinity ($HCO_3^-$) concentrations were measured within 12 hours of sampling by titration.

Major ion concentrations were measured at Monash University. Cation concentrations were determined on filtered (0.45 µm cellulose nitrate filters) water samples that were acidified to pH <2 with double distilled 16 N $HNO_3$ using ICP-OES (Thermo Scientific iCAP 7000). Concentrations of anions were determined on unacidified filtered water samples by ion chromatography (Thermo Scientific Dionex ICS-1100). Based on replicate analyses, the

precision of cation and anion concentrations are ±2 %; from the analysis of certified standards, accuracy is estimated at ±5 %. Total dissolved solids (TDS) concentrations are the sum of the of cation and anion concentrations.

$^3$H and $^{14}$C activities were measured at the Institute of Geological and Nuclear Sciences (GNS) in New Zealand. Samples for $^3$H analysis were vacuum distilled and electrolytically enriched

and $^3$H activities were measured by liquid scintillation as described by Morgenstern and Taylor (2009). Quantification limits are 0.02 TU are typical relative uncertainties are ±2 % (Table S1). $^{14}$C activities were measured by AMS following Stewart et al. (2004). Dissolved inorganic carbon (DIC) was converted to $CO_2$ by acidification with $H_3PO_4$ in a closed evacuated environment. The $CO_2$ was purified cryogenically and converted to graphite. $^{14}$C activities are

normalised using the $\delta^{13}$C values and expressed as percent modern carbon (pMC), where the $^{14}$C activity of modern carbon is 95 % of the $^{14}$C activity of the NBS oxalic acid standard in 1950. Uncertainties are between 0.27 and 0.35 pMC (Table S1).

### 3.3 Recharge calculations

Recharge rates were estimated using the methods discussed above. Recharge estimates from

the CMB (Eq. 1) utilised present-day average rainfall amounts (~640 mm) and Cl concentrations of 2.2 to 4.4 mg L$^{-1}$ together with the measured Cl concentrations of groundwater (Table S1). Recharge rates were estimated using the WTF method (Eq. 2) from the hydrographs of bores that display seasonal variations in water levels (Fig. 2). There is a





single pronounced annual increase in the hydraulic head following winter rainfall, and $\Delta h$ was

estimated as the difference between the highest head value and the extrapolated antecedent

recession curve, which is an estimate of the trace that the bore hydrograph would have followed

in the absence of recharge (Healy and Cook, 2002). Adelana et al. (2014) and Dean et al. (2015)

estimated that $S_y$ was between 0.03 and 0.1, which is appropriate for silty clay to coarse-grained

sediments. The TRR calculations (Eq. 3) used the [3]H activities in Melbourne rainfall as the

input function (Tadros et al., 2014); the rainfall prior to the atmospheric nuclear tests was

assumed to have had the same [3]H activity as present-day rainfall. The annual average [3]H

activity of present-day rainfall in both Melbourne and Gatum is ~2.8 TU (Tadros et al., 2014;

Table S1). The mean porosity is 0.15 in the pasture and 0.10 in the forest (Adelana et al., 2014).

Estimates of the values of $b$ are discussed below.

**3.4 Mean residence times**

Mean residence times (MRTs) and the covariance of [3]H and [14]C activities in groundwater were

estimated via lumped parameter models (LPMs: Zuber and Maloszewski, 2001; Jurgen et al.,

2012). LPMs relate the [14]C activity of water at time $t$ ($C_{out}$) to the input of [14]C over time ($C_{in}$)

via the convolution integral

$$C_{out}(t) = \int_0^\infty qC_{in}(t - \tau_m)\, e^{-\lambda_c\tau_m} g(\tau_m) d\tau_m \qquad (5)$$

(Zuber and Maloszewski, 2001; Jurgen et al., 2012) where $q$ is the fraction of DIC derived from

rainfall or the soil zone, $(t - \tau_m)$ is the age of the water, $\tau_m$ is the mean residence time, $\lambda_c$ is the

decay constant for [14]C ($1.21 \times 10^{-4}$ year[-1]), and $g(\tau_m)$ is the system response function that

describes the distribution of residence times in the aquifer (described in detail by Maloszewski

and Zuber, 1982; Zuber and Maloszewski, 2001; Jurgens et al., 2012). [3]H activities may be

calculated from the input of [3]H over time in a similar way using the decay constant for [3]H of

0.0563 yr[-1]. Unlike [14]C, [3]H activities are not changed by reactions between the groundwater

and the aquifer matrix, hence the $q$ term is omitted.





There are several commonly used LPMs. The partial exponential model (PEM) may be applied

aquifers where only the deeper groundwater flow paths are sampled. The dimensionless PEM

ratio defines the ratio of the unsampled to sampled depths of the aquifer (Jurgens et al., 2012).

This study used PEM ratios of 0.05 to 0.5 that cover the ratios of the sample to unsampled

portions of the aquifers at Gatum. The dispersion model (DM) is derived from the one-

dimensional advection-dispersion transport equation and is applicable to a broad range of flow

systems (Maloszewski and Zuber, 1982; Zuber and Maloszewski, 2001; Jurgens et al., 2012).

The dimensionless dispersion parameter (DP) in this model describes the relative contributions

of dispersion and advection. For flow systems of a few hundreds of metres to a few kilometres,

DP values are likely to be in the range of 0.05 to 1.0 (Zuber and Maloszewski, 2001). Other

commonly applied LPMs, such as the exponential-piston flow model, produce similar

estimates of residence times (Jurgens et al., 2012; Howcroft et al., 2017). The long-term

variation of atmospheric $^{14}$C concentrations in the southern hemisphere (Hua and Barbetti,

2004; McCormac et al., 2004) was used as the $^{14}$C input function, and the $^{3}$H activities in

rainfall for Melbourne (Tadros et al., 2014) was used as the $^{3}$H input function.

## 4. Results

**4.1 Hydraulic heads and properties**

The hydraulic heads in regional groundwater from both pasture and forest catchments decrease

from the upper to lower slopes implying that the regional groundwater flows southwards (Fig.

1b). In the pasture, the hydraulic heads in groundwater from all bores generally gradually

increase over several weeks to months following the onset of winter rainfall (Fig. 2). The

increase in hydraulic heads was higher in 2016, which was a year of higher than average rainfall

(~800 mm: Bureau of Meteorology, 2020). This was especially evident at bore 63 (Fig. 2). In

the forest, groundwater heads from bores in the upper (3663 and 3665) and mid (3668) slopes

decline uniformly over the monitoring period, and the groundwater head from bore 3658 near



the drainage zones does not show seasonal variations (Fig. 2). However, fluctuations of heads

from three bores near the drainage zones (3669) and lower slopes (3656 and 3657) show

seasonal variations similar to that of the groundwater in the pasture.

Values of $K_s$ range from 0.06 to 0.31 m day$^{-1}$ in the pasture and from 0.002 to 0.18 m day$^{-1}$ in

the forest catchments. The aquifers in the upper and lower slopes of pasture catchment have

the highest $K_s$ values of ~0.31 m day$^{-1}$, whereas $K_s$ values of the aquifers in the forest are lowest

on the lower slopes (Table S1). The aquifers contain rocks from the same stratigraphic unit,

and the heterogeneous hydraulic properties probably reflect the degree of weathering,

cementation, and clay contents.

## 4.2 Major ions

TDS concentrations of regional groundwater range from 282 to 7850 mg L$^{-1}$ in the pasture

catchment and 1190 to 7070 mg L$^{-1}$ in the forest catchment (Table S1); the lowest salinity

regional groundwater is from the upper slope of the pasture catchment. The TDS concentrations

of the shallow riparian groundwater (≤1 m depth) are between 3890 and 8180 mg L$^{-1}$ in the

pasture and from 169 to 13600 mg L$^{-1}$ in the forest (Table S1). The regional and riparian

groundwater from both catchments has similar geochemistry. Na constitutes up to 67 % of the

total cations on a molar basis, and Cl accounts for up to 91 % of total anions on a molar basis.

Cl concentrations range between 45.2 and 8140 mg L$^{-1}$, which significantly exceed the mean

concentrations of Cl in local rainfall (2.2 to 4.4 mg L$^{-1}$: Hutton and Leslie, 1958; Bormann,

2004; Dean et al., 2014). Molar Cl/Br ratios are between 180 and 884 with most between 450

and 830 (Fig. 3a), which spans those of seawater and coastal rainfall (~650: Davies et al., 1998,

2001). The observation that the Cl/Br ratios are significantly lower those that would result from

halite dissolution ($10^4$ to $10^5$: Kloppmann et al., 2001; Cartwright et al., 2004, 2006) and do not

indicates that Cl is predominantly derived from rainfall and concentrated by evapotranspiration.

As discussed by Herczeg et al. (2001), Cartwright et al. (2006), and Tweed et al. (2009),





amongst others, evapotranspiration rather than halite dissolution is the main process in

controlling groundwater salinity in southeast Australia. Ca and $HCO_3$ concentrations are

uncorrelated (Fig. 3b) indicating that the dissolution of calcite is not a major process

influencing groundwater geochemistry.

### 4.3 Radioisotopes

[3]H activities of the regional groundwater at Gatum of up to 1.48 TU (Table S1, Fig. 4), are

below the average annual [3]H activities of present-day rainfall in this region of ~2.8 TU (Tadros

et al., 2014; Table S1). The highest [3]H activities (>1 TU) are from the regional groundwater in

the upper slopes (15.5 m depth) and the drainage zone (~1.3 m depth) of the pasture catchment

and between 15.8 and 28.8 m depths in the forest catchment (Table S1). Regional groundwater

from ≥28 m depth in the lower slopes of the pasture catchment and the drainage zone of the

forest catchment locally have below detection (<0.02 TU) [3]H activities (Table S1). The [3]H

activities of the shallow riparian groundwater in the pasture vary from 0.26 to 0.79 TU with

the highest activities from the lower slopes (Table S1, Fig. 4). The riparian groundwater in the

forest catchment has [3]H activities ranging from 2.01 to 4.10 TU (Table S1, Fig. 4), which are

locally higher than the annual average [3]H activity of present-day rainfall (~2.8 TU). These high

[3]H activities probably reflect seasonal recharge by winter rainfall that in southeast Australia

has higher [3]H activities than the annual average (Tadros et al., 2014).

The [14]C activities in regional groundwater from the pasture and forest catchments range from

70.7 to 104 (pMC) and from 29.5 to 101 (pMC), respectively (Table S1, Fig. 4). The highest

[14]C activities (>100 pMC) are from the groundwater in the upper slopes of the pasture

catchment and the lower zones of the forest catchment that also has high [3]H activities (Table

S1). The lowest [14]C activities are from groundwater at 18 to 28.4 m depths in the mid-slope

and drainage lines of the forest catchment (Table S1). [14]C activities of the shallow riparian





groundwater are 85.5 to 102 pMC, with higher activities (>100 pMC) in the drainage zones of the forest catchment (Table S1, Fig. 4).

### 4.4 Mean residence times and mixing

The $^{3}$H and $^{14}$C activities help understand water mixing within the aquifers (Le Gal La Salle et al., 2001; Cartwright et al., 2006, 2013) and the mean residence times. The predicted $^{3}$H vs. $^{14}$C activities (Fig. 4) were calculated for all DIC being introduced by recharge ($q = 1$) and for 10% contribution of $^{14}$C-free DIC from the aquifer matrix ($q = 0.9$). The aquifers are dominated by siliceous rocks, and the major ion geochemistry implies little calcite dissolution. Similar values of $q$ were estimated for groundwater from siliceous aquifers elsewhere in southeast Australia (Cartwright et al., 2010, 2012; Atkinson et al., 2014; Raiber et al., 2015; Howcroft et al., 2017) and elsewhere (Vogel, 1970; Clark and Fritz, 1997). Much lower $q$ values are precluded as samples cannot lie to the right of the $^{3}$H vs. $^{14}$C curves (Cartwright et al., 2006, 2013).

Mixing between older (low $^{3}$H and low $^{14}$C) and recently-recharged groundwater (high $^{3}$H and high $^{14}$C) results in groundwater samples that plot the left of the decay trends in Fig. 4. It is difficult to calculate MRTs for these mixed waters; however, it is possible to estimate MRTs from the $^{14}$C activities for groundwater lying close to the predicted decay trends. The calculated MRTs are up to 3,930 years in the pasture and up to 24,700 years in the forest (Table 1, Fig. 5). Aside from the differences between the results of the different LPMs, there are uncertainties in $q$ and uncertainties in the input function of $^{14}$C. Nevertheless, the $^{14}$C activities imply that much of the regional groundwater have residence times of several thousands of years and was recharged prior to land clearing. These long residence times are consistent with the locally clay-rich nature of the aquifers and the moderate to low hydraulic conductivities.



### 4.5 Recharge rates

**4.5.1 Cl mass balance**

Recharge rates calculated from the CMB method (Eq. 1) using total rainfall of ~640 mm yr$^{-1}$ and Cl concentrations of 2.2 to 4.4 mg L$^{-1}$ are similar between the pasture (0.3 to 61.6 mm yr$^{-1}$) and forest (0.2 to 58.8 mm yr$^{-1}$) catchments (Fig. 6a). The typical recharge rates for most of the regional groundwater are from 0.3 to 2.5 mm yr$^{-1}$ in the pasture and 0.2 to 11.2 mm yr$^{-1}$ in the forest (Fig. 6a). The Cl/Br ratios imply that dissolution of halite is negligible (and no halite has been reported in the aquifers from the study area) and all the Cl is delivered by rainfall. Whether the rate of Cl delivery has been constant over long time periods is more difficult to assess; however, the rainfall Cl concentrations are typical of inland rainfall, and southeast Australia does not record major climate fluctuations such as glaciations or monsoons (Davies and Crosbie, 2018). The CMB technique also assumes that the export of Cl by surface runoff is negligible. The streams at Gatum discharge ~8 % of local rainfall, and while they locally have high salinities, some of the solutes that they export represents groundwater discharging into the stream (Adelana et al., 2014). If some direct export of Cl has occurred, the recharge estimates would be slightly lower than estimated above.

Because Cl in groundwater accumulates over hundreds to thousands of years (Scanlon et al., 2002, 2006), the CMB method generally yields longer-term recharge rates that, in Australia, largely reflect those pre-land clearing recharge (Alison and Hughes, 1978; Cartwright et al., 2007; Dean et al., 2015; Perveen, 2016). This can be demonstrated by mass balance (Cartwright et al., 2007). A 10 to 20 m thickness of aquifer with a unit area of 1 m$^2$ and porosity 0.03 to 0.1 contains 300 to 2000 L of water. Cl concentrations in the groundwater at Gatum range from 45.2 to 8140 mg L$^{-1}$ which equates to 1.4 x 10$^4$ to 1.6 x 10$^7$ mg Cl in that section of the aquifer. Annual rainfall of ~640 mm with Cl concentrations of 2.2 to 4.4 mg L$^{-1}$ would deliver 1410 to 2820 mg Cl per m$^2$ each year. Thus, it takes up to 11,500 years to deliver the Cl contained in





that section of the aquifer. This conclusion is also consistent with the long [14]C residence times of much of the deeper regional groundwater at Gatum. The higher recharge rates (25.3 to 61.6 mm yr[-1]) are from regional groundwater in the upper slopes of the pasture (bore 63) and from shallow riparian groundwater in the drainage zones (piezometer FD2) and lower slopes (piezometer FB1) of the forest (Fig. 6a). The groundwater at these sites has high [3]H and [14]C

activities, and the recharge rates from the CMB technique are thus likely to be representative of recent recharge.

### 4.5.2 Water table fluctuations

The recharge rates calculated using the WTF method (Eq. 2) from the bore hydrographs which show seasonal head variations assuming $S_y$ values of 0.03 to 0.1 (Adelana et al., 2014; Dean et

al., 2015). The estimated recharge rates range from 15 to 500 mm yr[-1] (2 to 78 % of rainfall) in the pasture and 30 to 400 mm yr[-1] (5 to 63 % of rainfall) in the forest (Fig. 6b). As with the CMB estimates, the recharge rates are generally high at the upper slopes of the pasture catchment (Fig. 6a). The WTF method requires the hydrograph recession curves to be estimated. There are significant steep and straight recession curves in the bore hydrographs (Fig. 2) that

can lead to errors in recharge estimates. The values of $S_y$ are not well known, which also results in uncertainties in the recharge estimates. Also, as discussed above, the presence of moisture in the unsaturated zone and capillary fringe may reduce the effective values of $S_y$ leading to recharge rates being overestimated. These uncertainties are discussed further below. Overall, the recharge rates estimated by this method are higher than those estimated using CMB and

reflect present-day recharge rates.

### 4.5.3 [3]H renewal rate

The recharge rates for bores and shallow piezometers were estimated using the [3]H activities and the TRR method (Eq. 3 and 4). These recharge rates were calculated for those groundwater samples which do not show the mixing of recent and older groundwater (Fig. 4). Regional





groundwater from nested bores commonly has different TDS contents, EC values, $^3$H and $^{14}$C

concentrations (Table S1), indicating that the groundwater is stratified. Much of the deeper

groundwater has low $^3$H and $^{14}$C activities implying that it is not recently recharged. Based on

these differences in geochemistry, $b$ is estimated as being between 1 and 5 m (Table S1). $b$

values for the shallow riparian groundwater are estimated as 1 to 2 m, which is the approximate

thickness of the shallow perched aquifers (Brouwer and Fitzpatrick, 2002). $n$ values of 0.03 to

0.1 (Adelana et al., 2014) were used. These values are appropriate for silty clay to coarser

aquifer lithologies in this area and are similar to the values of $S_y$.

Recharge rates from the regional groundwater are 0.5 to 14.0 mm yr$^{-1}$ in the pasture and 0.01

to 59.5 mm yr$^{-1}$ in the forest catchment with most in the range of 0.01 to 0.6 mm yr$^{-1}$ (Fig. 6c).

The higher recharge rates were from the upslopes of the pasture (14.0 mm yr$^{-1}$) and lower

slopes of the forest (59.5 mm yr$^{-1}$). The recharge rates in the riparian groundwater are from

0.05 to 0.5 mm yr$^{-1}$ in the pasture and 13.3 to 89.0 mm yr$^{-1}$ in the forest (Fig. 6c).

The average annual $^3$H activity in present-day rainfall at Gatum (~2.8 TU) is within the

predicted range of the $^3$H activities in present-day Melbourne rainfall (3.0 ± 0.2 TU), implying

that the Melbourne $^3$H input function is appropriate to use for this area. Assuming uncertainty

in the $^3$H input function of 5 to 10% (which is similar to the present-day variability of $^3$H

activities reported by Tadros et al., 2014) results in <5% uncertainties in recharge estimates.

The variation resulting from analytical uncertainties are lower than this. Recharge rates are

sensitive to the $b$ values, which are not explicitly known and may be variable. However, $b$ is

unlikely to be >5 m based on the observed degree of chemical stratification. The recharge rates

are again generally higher than those calculated using the CMB, which reflects the effects of

the initial land clearing. However, despite both reflecting post land clearing recharge, they are

significantly lower than those estimated using the WTF.





## 5. Discussion

As expected, the recharge estimates from the CMB method are generally lower than those from the WTF and TRR methods reflecting the increase in recharge caused by the initial replacement of native eucalyptus vegetation by pasture. Although both the methods determine present-day recharge rates (Scanlon et al., 2002, 2006), recharge rates estimated using the WTF method are significantly higher than the TRR estimates (Fig. 7). Some differences will result from

uncertainties in $S_y$, $b$ and $n$. Because the values of $S_y$ and $n$ are likely to be similar, modifying their values would not resolve the issue of the large mismatch between the recharge rates estimated with the two methods. The value of $b$ would have to be increased up to 50 m to achieve agreement between the recharge estimates from the TRR and the WTF methods. This is not possible given the observations that groundwater major ions geochemistry and $^3$H and

$^{14}$C activities vary over vertical scales of a few metres (Table S1), implying that the groundwater is compartmentalised on those scales. It is also unlikely given the heterogeneous nature of the aquifers, which comprise interlayered clays and silts.

The highest recharge rates from the WTF method are >50% of rainfall. Such high recharge rates are unlikely given that evapotranspiration rates in this region are estimated to approach

the rainfall rates (Dean et al., 2016; Dresel et al., 2018; Azarnivand et al., 2020). The lower limits of the recharge rates estimated from the WTF method appear more reasonable but are still larger than most recharge rates estimated from the TRR method. The observation that much of the older saline groundwater has not been flushed from the catchments also implies that present-day recharge rates cannot be very high. Relatively high WTF recharge rates (up to 161

and 366 mm yr$^{-1}$) were also calculated in adjacent catchments with similar land-use (Dean et al., 2015; Perveen, 2016). $^3$H activities in groundwater from those catchments are similar to those in the same region, implying that recharge estimates based on the TRR method would again be significantly lower. Cartwright et al. (2007) and Crosbie et al. (2010) also reported





that the recharge estimates from the TRR method in semi-arid catchments elsewhere in

Australia are lower than those from the WTF method.

There are several possible reasons that might explain why the WTF method may overestimate

recharge. Air entrapped during recharge may increase the pressure in the aquifer (the Lisse

effect: Krul and Liefrinck, 1946; Meyboom, 1967; McWhorter, 1971). However, this occurs

during rapid recharge, which is not observed in the Gatum area. Dean et al. (2015) suggested

that the high recharge rates estimated from the WTF method in the adjacent Mirranatwa

catchments might reflect focussed recharge from streams. This is not the case at Gatum as high

WTF recharge rates are recorded at all landscape positions and the streams only export ~8% of

rainfall (Adelana et al., 2014). Because the WTF estimates gross recharge and geochemical

methods estimate net recharge, there may be differences if the water is removed from the water

table by evapotranspiration. The plantation forest plausibly has high evapotranspiration rates

(Benyon et al., 2006; Dean et al., 2015; Dresel et al., 2018); however, this explanation is

unlikely in the pasture where water tables are locally several metres below land surface, and

there is not deep-rooted vegetation.

It is most likely that the higher recharge rates estimated from the WTF method reflect the lower

effective $S_y$ caused by moisture in the unsaturated zone and the capillary fringe (Gillham, 1984;

Crosbie et al., 2005, 2019). This conclusion is consistent with the soils being fine-grained and

thus likely to retain moisture between recharge events. While it is difficult to test this possibility,

it is clear that the estimates from the WTF method are higher than expected, given the

evapotranspiration rates and the preservation of old groundwater in these catchments. The

recharge rates estimated from the TRR method are still subject to uncertainty (especially in

determining $b$) but are probably a more reasonable estimate.



### 5.1. Understanding the impacts of reforestation

The recharge estimates from the TRR method differ little between the pasture and the forest, which is unexpected given that the establishment of plantation forests aimed to reduce the

recharge rates. The evapotranspiration rates in the forest are also higher than in the pasture (Adelena et al., 2014; Dresel et al., 2018) and water levels are declining in some areas of the forest with no corresponding decline in the pasture (Fig. 2), which suggests higher water use by the trees. The plantation covers ~62% of the forest catchment, and many of the bores are in cleared areas between the stands of trees (Fig. 1a). Thus, the recharge rates may not be

representative of the forest as a whole. Additionally, the TRR averages recharge rates over the timespan of the residence times of the aliquots of water contained in the water sample, which may be several years to decades (Maloszewski and Zuber, 1982; Cartwright et al., 2017). Thus, the recharge rates in the forest catchment may reflect those from both before and following the recent reforestation.

## 6. Conclusions

Estimating recharge rates is fundamentally important to assess the impacts of land clearing and subsequent reforestation in semi-arid areas. As has been discussed elsewhere (Scanlon et al., 2002; Healy, 2010), using a range of techniques together with other data (such as estimates of residence times) is required to fully understand recharge. The groundwater geochemistry and

hydraulic heads can discern the broad increases to recharge caused by the initial replacement of native eucalypt forest by pasture. However, probably due to a combination of bore location and the averaging of recharge rates by geochemical methods over several years, any changes to recharge caused by the recent establishment of plantation forests are less obvious. While the WTF method should be able to estimate recharge on shorter timescales, there may be problems

in its application due to the influences of moisture in the unsaturated zone on the effective $S_y$.



Additionally, the recharge rates are spatially variable across both catchments, and even with a relatively high density of data, it is difficult to estimate typical or area-integrated values.

An accurate estimation of recharge rates is important for groundwater modelling, because recharge represents the water flux used as a boundary condition at the water table. Integrated
surface and subsurface hydrologic models, which usually simulate the coupled surface and soil water fluxes accounting for both the unsaturated and saturated zones, require, in catchments with intermittent streamflow, groundwater observations for calibration of parameters and evaluation of model results. Considering the uncertainties associated with different experimental methods, integrated surface and subsurface models might represent a valuable
alternative to quantify recharge rates, with also possible applications to solute and isotope transports (Scudeler et al., 2016; Daneshmand et al., 2019).

As previous studies have shown, present-day recharge rates in the pasture, which is typical of cleared land in southeast Australia, are generally <10 mm yr$^{-1}$. Despite these being significantly higher than the pre-land clearing recharge rates, they only result in the gradual replacement of
the older saline water stored in these aquifers. Thus, significant freshening of the saline groundwater that exists over much of southeast Australia will only occur over extended periods. Additionally, while there has been a rise in the water table caused by the increased recharge, and in some cases increased drainage in the streams, the magnitude of these changes will be limited by the modest recharge rates.

*Author contribution:* Shovon Barua and Ian Cartwright conducted the sampling assisted by P. Evan Dresel and Edoardo Daly. Shovon Barua carried out the analytical work conducted at Monash University. P. Evan Dresel and Edoardo Daly manage the field site and provided pre-existing data. All authors were involved in writing the manuscript.

*Competing interests:* The authors declare that they have no conflict of interest.



*Acknowledgements:* We thank Rob Lawrence (farm owner) and Georgie Luckock (plantation manager at PF Olsen Australia) for allowing us to access their farm and forest. Special thanks to Peter Hekmeijer from the Department of Jobs, Precincts and Regions, Victoria for his keen support in this study. Dr. Massimo Raveggi and Mrs. Rachelle Pierson are thanked for their help with the analytical work. This project was funded by the Australian Research Council

through its Discovery Program (grant DP180101229).

*Data Availability:* All analytical data is presented in the Supplement. Groundwater head data are from Dresel et al. (2018).

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

## Figures

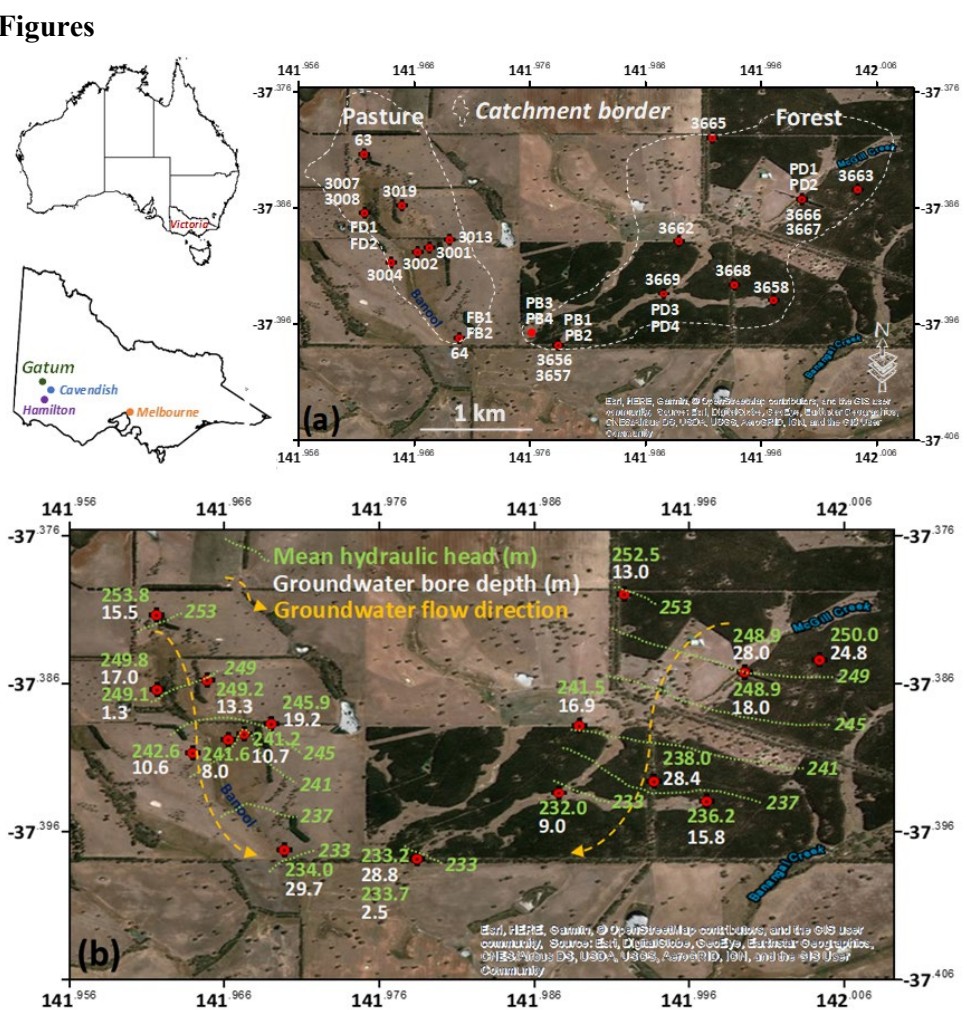

Figure 1: (a) Map of the Gatum pasture and forest catchments with the locations of groundwater
bores (3007 & 3008, 3666 & 3667, and 3656 & 3657 are nested bores); shallow piezometers
are at PD (pasture drainage zone), PB (pasture lower slope), FD (forest drainage zone), and FB
(forest lower slope). The catchment border is from Dresel et al. (2018). (b) Mean hydraulic
heads of groundwater from 2010 to 2017 except for 3008 (from 2010 to 2015) and 3658 (from
2010 to 2016) with bore depths and flow directions. Background ArcGIS®10.5 image (Esri,
HERE, Garmin, ©OpenStreetMap contributors and the GIS User Community, Source: Esri,
DigitalGlobe, GeoEye, Earthstar Geographics, CNESAirbus DS, USDA, USGS, AeroGRID,
IGN, and the GIS User Community). © OpenStreetMap contributors 2020. Distributed under a
Creative Commons BY-SA License



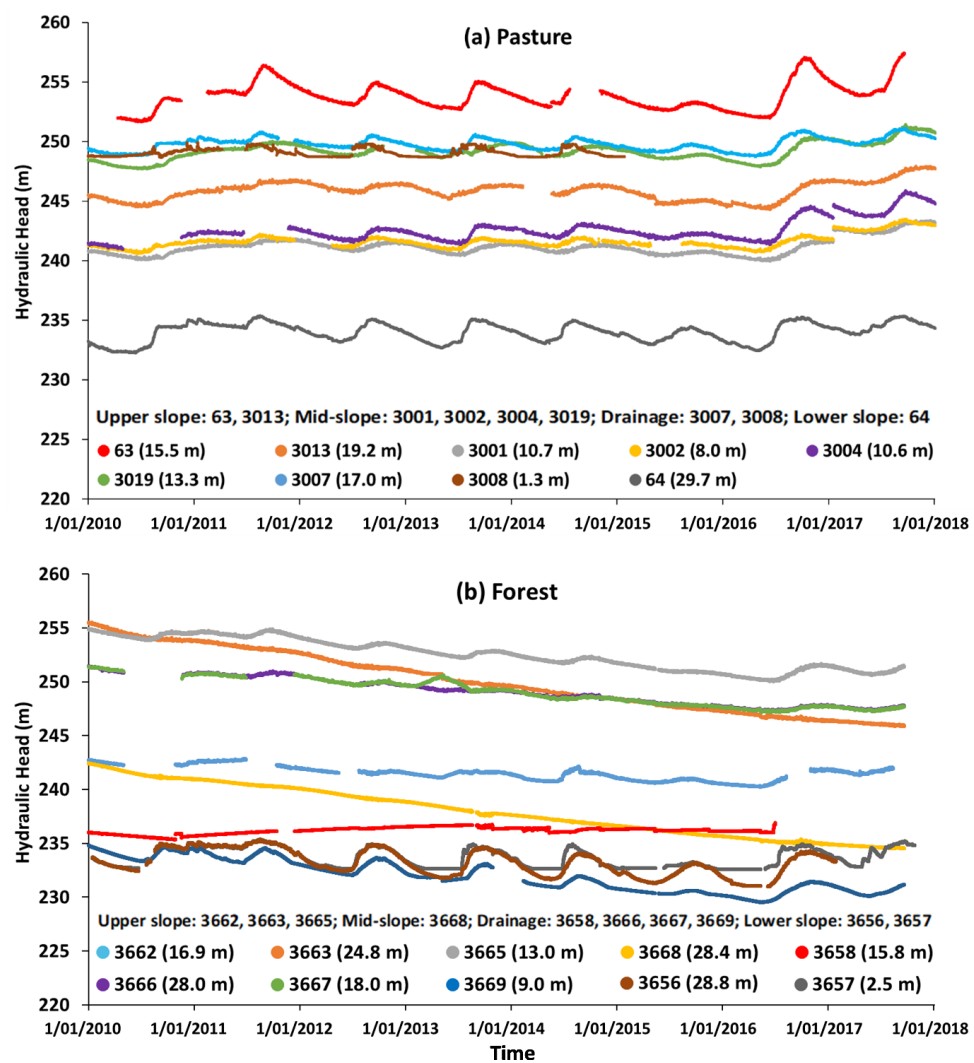

865

Figure 2: Variation in groundwater heads from bores in (a) pasture and (b) forest (Dresel et al., 2018). The legend shows the screen depths (numbers in parentheses) and landscape positions.



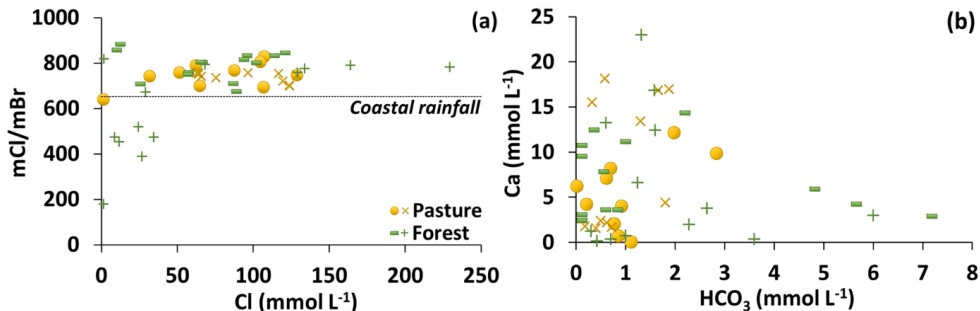

Figure 3: (a) variation in molar Cl/Br ratios with molar concentrations of Cl, (b) molar Ca vs. HCO₃ concentrations. Cross and plus symbols are for shallow riparian groundwater other symbols are for regional groundwater.

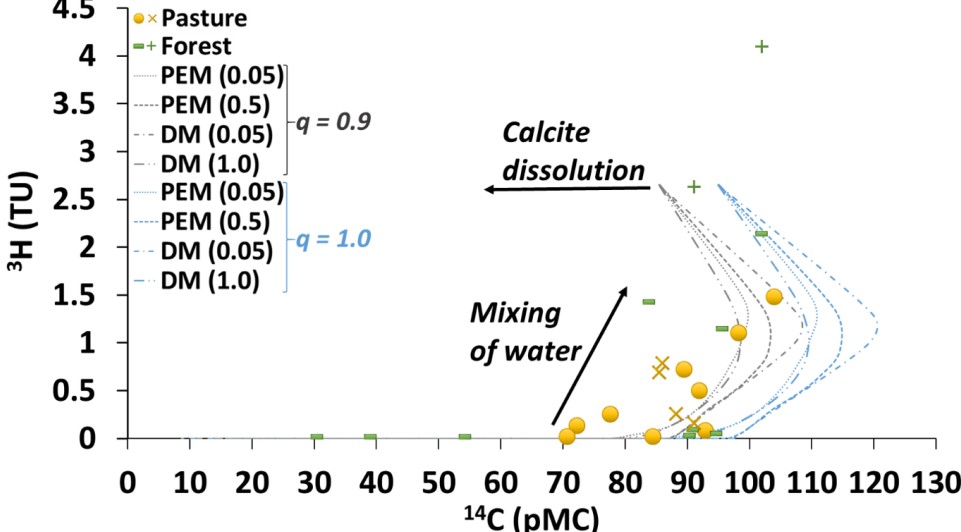

Figure 4: The activities of $^3$H (TU) and $^{14}$C (pMC) in pasture and forest groundwater. PEM = partial exponential model (PEM ratio in brackets) and DM = dispersion model (DP parameter in brackets). Cross and plus symbols are for shallow riparian groundwater other symbols are for regional groundwater. The single high $^3$H activity possibly reflects seasonal recharge. Samples lying off the covariance curves probably record mixing between younger and older groundwater.

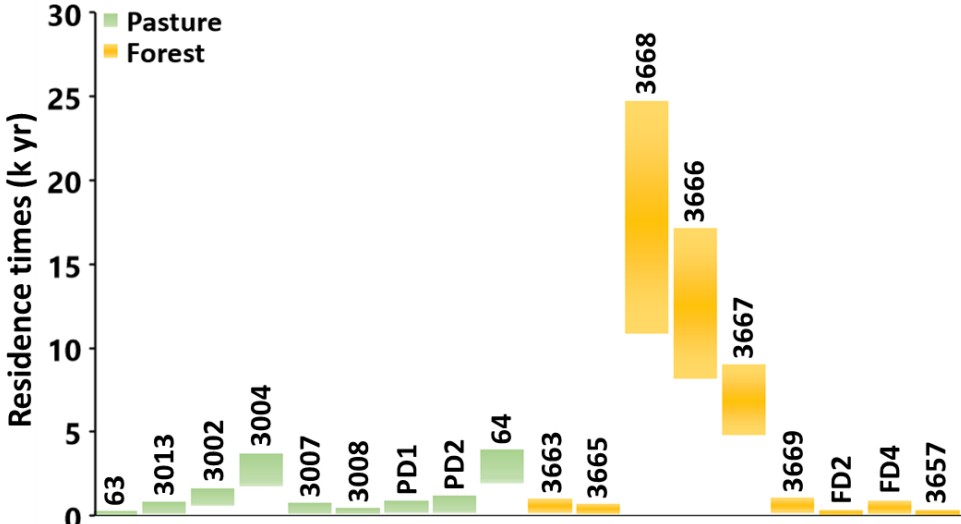

Figure 5: The ranges of groundwater residence times in ka estimated using different LPMs. The numbers above the box represent sample IDs. PD and FD represent the shallow groundwater in the pasture drainage and forest drainage areas, respectively. Data from Table 885 1.



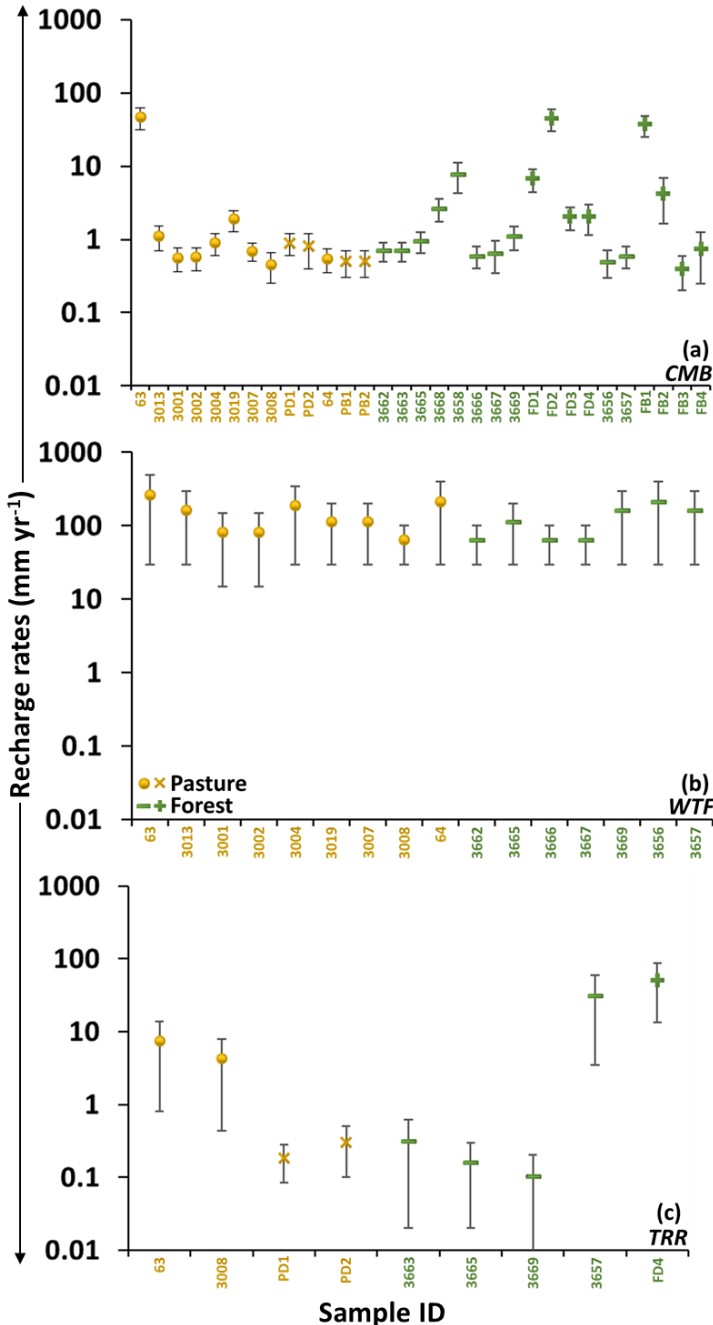

Figure 6: Recharge rates in mm yr$^{-1}$ estimated from (a) CMB, (b) WTF and (c) TRR. PD and FD are for shallow groundwater in pasture drainage and forest drainage areas, respectively.

890    Bars indicate the ranges of recharge rates from Table 1.

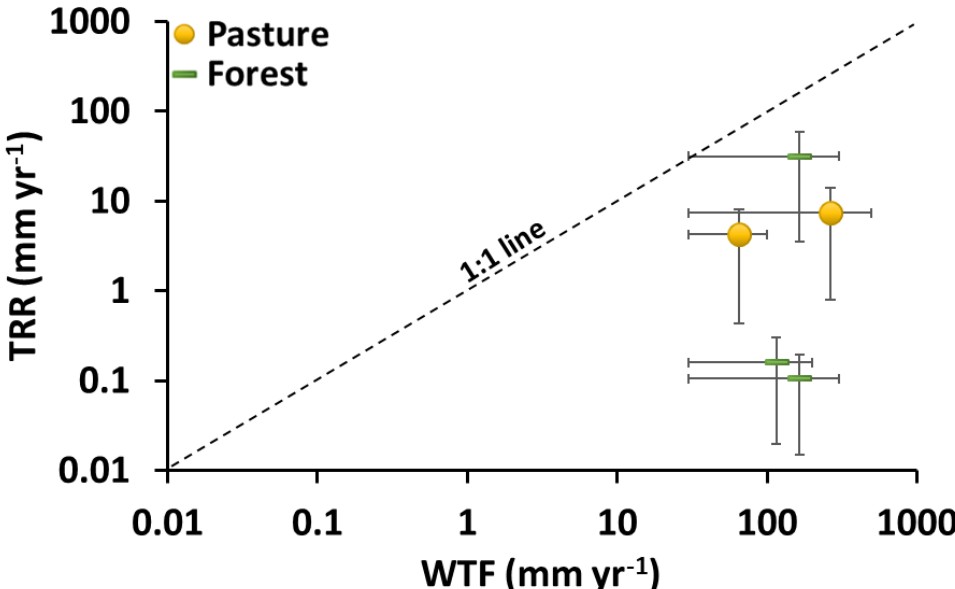

Figure 7: Comparison between recharge rates for regional groundwater estimated from WTF and TRR. Bars represent the ranges of calculated recharge values from Table 1.





895    Table 1: Groundwater recharge rates and estimated residence times of groundwater.

| Sample | Sample depth (m) | Landscape position | Recharge rates (mm yr$^{-1}$) | | | Groundwater residence times (yr) | | | | | | | |
|---|---|---|---|---|---|---|---|---|---|---|---|---|---|
| | | | WTF | CMB | TRR | PEM (0.05) | | PEM (0.5) | | DM (0.05) | | DM (1.0) | |
| | | | | | | q = 0.9 | q = 1.0 | q = 0.9 | q = 1.0 | q = 0.9 | q = 1.0 | q = 0.9 | q = 1.0 |
| *Pasture Catchment* | | | | | | | | | | | | | |
| 63 | 15.5 | Upper | 30-500 | 31.7-61.6 | 0.8-14.0 | | 180 | 60 | 150 | 70 | 80 | | 270 |
| 3013 | 19.2 | Upper | 30-300 | 0.7-1.5 | | 210 | 780 | 140 | 690 | 90 | 680 | 270 | 780 |
| 3001 | 10.7 | Mid | 15-150 | 0.4-0.7 | | | | | | | | | |
| 3002 | 8 | Mid | 15-150 | 0.4-0.7 | | 660 | 1470 | 540 | 1380 | 540 | 1290 | 650 | 1620 |
| 3004 | 10.6 | Mid | 30-350 | 0.6-1.2 | | 2010 | 3200 | 1860 | 2910 | 1710 | 2730 | 2220 | 3650 |
| 3019 | 13.3 | Mid | 30-200 | 1.3-2.5 | | | | | | | | | |
| 3007 | 17 | Drainage | 30-200 | 0.5-0.9 | | 190 | 720 | 160 | 600 | 90 | 600 | 270 | 720 |
| 3008 | 1.3 | Drainage | 30-100 | 0.3-0.6 | 0.5-8.0 | 70 | 390 | 110 | 200 | 80 | 120 | 90 | 420 |
| PD1 | 1 | Drainage | | 0.6-1.2 | 0.05-0.3 | 240 | 860 | 170 | 750 | 110 | 740 | 320 | 870 |
| PD2 | 1 | Drainage | | 0.4-1.2 | 0.08-0.5 | 390 | 1080 | 200 | 1020 | 120 | 960 | 420 | 1170 |
| 64 | 29.7 | Lower | 30-400 | 0.4-0.7 | | 2240 | 3470 | 2070 | 3150 | 1920 | 2960 | 2510 | 3930 |
| PB1 | 1 | Lower | | 0.3-0.7 | | | | | | | | | |
| PB2 | 1 | Lower | | 0.3-0.7 | | | | | | | | | |
| *Forest Catchment* | | | | | | | | | | | | | |
| 3662 | 16.9 | Upper | 30-100 | 0.5-0.9 | | | | | | | | | |
| 3663 | 24.8 | Upper | | 0.5-0.9 | 0.04-0.6 | 320 | 960 | 180 | 870 | 110 | 830 | 360 | 990 |
| 3665 | 13 | Upper | 30-200 | 0.6-1.3 | 0.02-0.3 | 170 | 660 | 150 | 540 | 90 | 560 | 250 | 660 |
| 3668 | 28.4 | Mid | | 1.8-3.5 | | 17000 | 19600 | 13100 | 14700 | 10800 | 11900 | 21400 | 24700 |
| 3658 | 15.8 | Drainage | | 4.3-11.2 | | | | | | | | | |
| 3666 | 28 | Drainage | 30-100 | 0.4-0.8 | | 11500 | 13600 | 9480 | 10900 | 8160 | 9230 | 14300 | 17100 |
| 3667 | 18 | Drainage | 30-100 | 0.4-0.9 | | 5850 | 7440 | 5160 | 6450 | 4780 | 5870 | 6930 | 9000 |
| 3669 | 9 | Drainage | 30-300 | 0.7-1.5 | 0.01-0.2 | 330 | 990 | 180 | 930 | 110 | 870 | 380 | 1020 |
| FD1 | 1 | Drainage | | 4.6-8.9 | | | | | | | | | |
| FD2 | 1 | Drainage | | 30.3-58.8 | $^3H$ (>2.8) | | 210 | 70 | 170 | 80 | 90 | | 300 |
| FD3 | 1 | Drainage | | 1.4-2.7 | | | | | | | | | |
| FD4 | 1 | Drainage | | 1.2-2.9 | 13.3-89.0 | 260 | 860 | 170 | 750 | 90 | 740 | 320 | 870 |
| 3656 | 28.8 | Lower | 30-400 | 0.3-0.7 | | | | | | | | | |
| 3657 | 2.5 | Lower | 30-300 | 0.4-0.8 | 3.6-59.5 | | 300 | 90 | 170 | 80 | 110 | | 330 |
| FB1 | 1 | Lower | | 25.3-49.0 | | | | | | | | | |
| FB2 | 1 | Lower | | 1.7-6.8 | | | | | | | | | |
| FB3 | 1 | Lower | | 0.2-0.6 | | | | | | | | | |
| FB4 | 1 | Lower | | 0.3-1.2 | | | | | | | | | |

Landscape positions: Upper, Mid, and Lower slopes and Drainage Zones as discussed in text. Sample depth is the middle of the screened interval. The recharge rates from WTF method were calculated for bore hydrographs that show seasonal variations in hydraulic head. The recharge rates with TRR were calculated assuming $b$ was 1 to 5 m (bores) and 1 to 2 m (shallow piezometers). Recharge rates from
900    TRR and residence times were only calculated for groundwater samples that do not show mixing of young and old groundwater. Groundwater residence times not calculated for samples with higher $^{14}C$ concentrations than in lumped parameter model outputs.