# Peer review of "Using multiple methods to investigate the effects of land-use changes on groundwater recharge in a semi-arid area"

_Hydrology and Earth System Sciences, 2020_

## Referee Comment (RC1) · Anonymous Referee #1 · 27 May 2020

This paper presents an interesting case study of three methods to estimate ground-water recharge in two small catchments subject to landuse change. Chloride mass balance (CMB), tritium renewal rate (TRR) with carbon-14 in a lumped parameter modelling approach were demonstrated as complimentary methods to estimate recharge. However a third method, the water table fluctuation (WTF) method resulted in large overestimates of recharge. The study compared recharge for a pasture catchment (151 ha) and forest catchment (338 ha), both drained by intermittent streams, at the head of the Glenelg river in western Victoria, Australia.

GENERAL COMMENTS

Although an interesting case study, the study provides limited justification and context, with some broad statements that should be better supported. How does this study

inform sustainable management of groundwater (from the opening line of the abstract)? The description of the study area does not mention groundwater use in lower in the catchment, or reference to estimates of sustainable yields on a larger scale. A context of groundwater management issues in the region is not provided. How do the authors reconcile their view of the importance of recharge estimates with the 'water budget myth'? A related myth that sustainable development of groundwater resources can be defined by groundwater residence times has recently been highlighted by Ferguson et al 2020, citing classic papers on the water budget myth. The paper is well written and presented, although some additional figures to provide context and explanation would be helpful. Specific suggestions are provided below.

SPECIFIC COMMENTS

A number of more specific queries and comments follows:

1) The objectives of study were to examine uncertainties in varying methods of estimating recharge. However, there is no discussion of how the method comparison is similar or distinct from other recharge studies in semi-arid areas. Have other studies also found the WTF method overestimates recharge for example?

2) Comparing methods for recharge rates is interesting, but the authors argue (Line 481) that it is 'fundamentally important to asses the impacts of land clearing'. Why?

3) Section 5.1 on impacts of reforestation only considers the TRR method, which surprisingly does not find significant difference in recharge between pasture and forest. Other evidence indicates the forest is using more water, so the study appears to demonstrate the limitations of recharge estimation methods?

4) How do the authors recommend these results inform groundwater modelling ? Line 495

5) Both WTF and TRR rely on estimating the effective porosity (or effective specific yield). Mean porosity was previously reported as 0.15 and 0.1 respectively for the

pasture and forested catchments, but is unclear how this was determined, and how sensitive the WTF and TRR methods are to the range of possible values. Line 385 states Sy is 'not well known' which is an understatement, as the parameter is highly uncertain. There is also a possibility of semi-confined conditions to develop at very shallow depths and that hydraulic loading could account for part of the water level response to rainfall.

6) CMB method is most reliant on assumption of long term rate of Cl delivery, and can only be applied in catchments with negligible runoff and sedimentary Cl inputs. How are the results sensitive to 8% runoff measurement from the catchments?

7) The limitations of lumped parameter models (LPMs) should be discussed, as the dimensionless ratios assumed vary over a very wide range (eg. 0.05 to 1). Are the estimated residence times linearly related to these lumped parameters? Also, can it be clarified why the PEM and DM lumped parameter models were applied and not the exponential-piston flow model?

8) Clarify Line 295, regarding Cl/Br ratios 'and do not indicates that Cl is predominantly derived from rainfall and concentrated by evapotranspiration'.

9) Schematic cross-sections could help explain the relationship between regional vs. riparian groundwater. An additional map that shows the regional catchment context of the catchment divides for groundwater vs. surface water would also be helpful, as the current mapping provides very large scale and small scale maps.

10) Mean residence times, estimated from both 3H and 14C , were ∼4K in pasture and ∼24K in forest. Yet the forest was planted only ∼20 years ago, after ∼160 years of pasture. The CMB method suggests chloride accumulation over ∼10K years of rainfall inputs, to account for relatively high salinities. These differential time scales should be discussed further.

Ferguson G, Cuthbert MO, Befus, K, Gleeson T, McIntosh J (2020).The groundwater

age-sustainability myth. https://eartharxiv.org/gq2m3/

---

## Referee Comment (RC2) · Anonymous Referee #2 · 1 Jun 2020

The authors present an interesting case study in a semi-arid area in Australia. They used three recharge estimation methods for two small catchments subject to landuse change. Based on the Chloride mass balance (CMB), Tritium renewal rate (TRR) and Water table fluctuation (WTF) method recharge rates were estimated. Moreover mixing between older and recently recharged groundwater as well as mean residence time was calculated. Recharge rates are relatively modest but the WTF overestimate recharge rates due to a simplified used of a constant Sy. A stratification was identified where the older groundwater can have a mean residence time of  $\sim$ 25000a and the thickness of the upper part of the aquifer system that receives recent recharge is less than 5 m.

Overall, the paper is well written and the results are solid. However, the most critical

point I see is that the study do not provides a broader context. How does the results impact water management for the study area and for the region. Regarding the uncertainty in the estimated recharge rates and spatial and temporal variability, it is not obvious for me how sustainable water resource management can set-up. Perhaps the authors have some thoughts about this problem and might provide some suggestions. Moreover, if one of the objectives of this study is to assess and compare uncertainty in the methods, then this has to be more elaborated and systematically compared. In addition, these results should be compared to similar studies. Furthermore, I miss a conceptual model which describe the processes. That can be a schematic figure or a cross-section describing the different flow systems and geochemical signatures.

Some further comments and suggestions are provided below. Introduction: Personally, I believe that the study objectives should be clearly communicated in 1. Introduction. I found it a bit confusing to get information about the different methods before knowing the target of the study. Line 48ff. Not only in semi-arid areas recharge varies in space and time. Also in humid areas recharge can be considerable spatially and temporally different (see for example, Moeck et al., 2020 and Mohan et al., 2018, among many others) Line 50ff: You could add Darcy methods, soil moisture methods, heat tracers, baseflow separation techniques, empirical relationships, etc. for completeness of the provided list (see for instance Healy 2010, Walker et al., 2019). Section 1.1.1. When residence times are around ~25000 years, how likely is that all CI is originating from rainfall only and the impact of runoff can be neglected. This is more a questions rather than a critic. You already indicate based CI/Br ratios that evapotranspiration rather than halite dissolution is the main process in controlling groundwater salinity but would could be the error in estimated recharge rates if a small amount of Cl is not only originating from precipitation? Line 85ff: In the study area with an actual ET of  $\sim$ 600 mm/a, to what depth can ET impacts be observed. I am asking because I am not sure if the observation wells 3008 (depth 1.3, pasture) and 3657 (depth 2.5 m, forest with deeper root zones) can be reliable used by applying the water table fluctuation method, although I have to note that the estimated rates seems to be in the same range like for HESSD
the other observation points. Line 295-297: Maybe I misunderstood something here, but did you not indicate that all CI is delivered by rainfall (e.g. Line351). Please check the statement and maybe reformulate the second part of the sentence. Line333ff: Not clear. Please explain why it is not possible. Line392ff: Just from Fig.4 it is not possible to identify the samples. Perhaps you can better highlight these samples in Fig. 4 or provide the link to Table 1. Apart from that I am wondering that location 3663 do not show mixing with older groundwater, even though it is one of the deepest location ( $\sim$ 25m) and based on the drawn picture with the stratification I was expecting that older groundwater exist. Moreover, the decline in the groundwater level for 3663 is uniformly over the monitoring period, which is in contrast to the wells 3657 and 3669. But for all these the TRR is applied and no mixing is assumed. Could you please elaborate more on these differences? Line 491ff: Yes, I absolutely agree and this is an important point. The question which arise from the results is how we can set-up sustainable water resource management then when we have such a spatial and temporal variability as well as uncertainty on  $\sim$ 500 ha. Perhaps the authors have some thoughts about this problem and might provide some suggestions. Line 495: Although I agree that physical based models are useful tools, the models will likely not represent in every detail the recharge processes because of the lack of observations although a relatively high density of data exist and information about the subsurface heterogeneity are missing (in both, xy and z direction). Thus, calibration is required which will also lead to uncertainty. For me, this part sounds like we always should use physical based models and then we get the right recharge rates which is obviously not true, even though the models are very powerful. Please reformulate. Figure 1: If available, it would be useful to add the precipitation on top of the graphics (second y-axis). Also why does well 3658 show an increase?

Healy, R.W., 2010. Estimating Groundwater Recharge. Cambridge University Press.

Moeck, C., Grech-Cumbo, N., Podgorski, J., Bretzler, A., Gurdak, J. J., Berg, M., & Schirmer, M. (2020). A global-scale dataset of direct natural groundwater recharge
rates: A review of variables, processes and relationships. Science of The Total Environment, 717, 137042.

Mohan, C., Wei, Y., & Saft, M. (2018). Predicting groundwater recharge for varying land cover and climate conditions–a global meta-study. Hydrology and Earth System Sciences, 22(5), 2689-2703.

Walker, D., Parkin, G., Schmitter, P., Gowing, J., Tilahun, S. A., Haile, A. T., & Yimam, A. Y. (2019). Insights from a multi-method recharge estimation comparison study. Groundwater, 57(2), 245-258.

---

## Short Comment (SC1) · 19 Jun 2020

We thank the reviewer for their helpful comments, our responses are below (in blue)

Although an interesting case study, the study provides limited justification and context, with some broad statements that should be better supported. How does this study inform the sustainable management of groundwater (from the opening line of the abstract)? The description of the study area does not mention groundwater use lower in the catchment, or reference to estimate of sustainable yields on a larger scale. A context of groundwater management issues in the region is not provided.

This was also raised by Reviewer #2. The study took place at catchments instrumented to study the effects of land-use change on groundwater and surface water resources, providing infrastructure and a data record needed for this investigation. The aims of our study were a better understanding of the uncertainties and limitations of commonly-applied recharge methods using these well-instrumented catchments as examples. As we explained, estimating recharge is important to understand the hydrogeology of semi-arid regions and this study has relevance to other regions where these techniques are used. Groundwater is not extensively used in this catchment, and our comments on sustainability were a more general recognition that groundwater can be a vital resource in semi-arid areas. In terms of the specific need to understand recharge in these catchments, defining whether recharge rates changed with land-use changes is important (especially the potential impacts on waterlogging and streamflow caused by changing water table elevations). We will clarify the aims and implications of the revised manuscript.

How do the authors reconcile their view of the importance of recharge estimates with the 'water budget myth'? A related myth that sustainable development of groundwater resources can be defined by groundwater residence times has recently been highlighted by Ferguson et al. 2020, citing classic papers on the water budget myth.

This is beyond what we can easily discuss in this paper. We agree that defining sustainable yields is difficult; however, understanding recharge is a critical part of assessing the impacts of groundwater use (e.g., Gleeson et al., 2012, Nature, 11295). The use of "residence times" also differs between this paper and Ferguson et al. (2020) study. We use it to refer to individual samples rather than the average age of water in the aquifer as a whole (sometimes called the turnover time). We avoid using the term "age" for specific samples as it is not a valid concept for most groundwater (Suckow et al., 2014, Applied Geochemistry, 50, 222–230). Given that the paper does not discuss sustainability in detail, we will reword the introduction and abstract to better reflect this.

The paper is well written and presented, although some additional figures to provide context and explanation would be helpful.

Specific suggestions are provided below.

1) The objectives of the study were to examine uncertainties in varying methods of estimating recharge. However, there is no discussion of how the method comparison is similar or distinct from other recharge studies in semi-arid areas. Have other studies also found the WTF method overestimates recharge for example?

We discussed this at lines 439-441 with reference to previous studies (Cartwright et al., 2007, Journal of Hydrology, 332, 69-92; Crosbie et al., 2010, Hydrology and Earth System Sciences, 14, 2023-2038; Dean et al., 2015, Hydrology and Earth System Sciences, 19, 1107–1123; Perveen, 2016, http://hdl.handle.net/1959.9/560005). We will reinforce this in the discussion and also make this point more generally in the introduction. Several of these studies (including the review of Crosbie et

al., 2010, Hydrology and Earth System Sciences, 14, 2023-2038, and Crosbie et al., 2019, Water Resources Research, 55, 7343-7361) show that the WTF method generally overestimates recharge rates (lines 443-444), mainly due to specific yields being overestimated. We will discuss this more fully in the revised manuscript.

2) Comparing methods for recharge rates is interesting, but the authors argue (Line 481) that it is 'fundamentally important to assess the impacts of land clearing'. Why?

Inevitably understanding the impacts of land clearing necessitates using different methods to understand pre- and post-land clearing recharge. Pre-land clearing recharge is commonly estimated using the CMB or longer-lived radioisotopes (e.g., $^{14}$C). Modern recharge may be estimated using the WTF method or Tritium. However, this assumes that the rates from those different methods are all broadly correct (or at least comparable), which is probably not the case. This is an important general point that we will emphasise in the introduction and discussion of the revised paper.

3) Section 5.1 on the impacts of reforestation only considers the TRR method, which surprisingly does not find a significant difference in recharge between pasture and forest. Other evidence indicates the forest is using more water, so the study appears to demonstrate the limitations of recharge estimation methods?

This is a good point. The area was partially replanted in the past 20 years. We discussed that the WTF method overestimates recharge rates and CMB method yields long-term recharge rates. $^{3}$H activities and the TRR method are applicable to understanding the initial land use changes. We will specify this in the revised paper. There is also a scale issue. Most recharge rates based on groundwater samples from bores average recharge over areas of a few 10's to 100's m$^2$ (Scanlon et al., 2002, Hydrogeology Journal, 10, 18-39), and commonly the bores in plantation forests are in cleared areas that may be larger than this, possibly underestimating the recharge rates. This was discussed on lines 473-475. The TRR method has a time lag of several years (see Comment 10) that means that it may not yet detect the latest reforestation. We will clarify this in the revision.

4) How do the authors recommend these results inform groundwater modelling? Line 495.

Popular groundwater models, such as MODFLOW, use recharge rates as a boundary condition at the water table. Isotope methods give estimates of recharge rates across large areas over a relatively long period of times dating back hundreds of years. Because the WTF method estimates recharge rates at smaller spatial scales for the years when data are available, it is often  considered to be more appropriate to constrain the models. However, the large discrepancies in the values of recharge rates from different techniques and the overestimated recharge rates with the WTF method pose questions on the quantification of boundary conditions for groundwater models. As suggested in our discussion, the use of integrated surface and subsurface hydrologic models might overcome this issue and provide an additional tool for the estimation of recharge rates to support experimental analyses. We will modify this statement in the manuscript to strengthen this part of the conclusions.

5) Both WTF and TRR rely on estimating the effective porosity (or effective specific yield). Mean porosity was previously reported as 0.15 and 0.1 respectively for the pasture and forested catchments but is unclear how this was determined, and how sensitive the WTF and TRR methods are to the range of possible values.

The values of porosity were taken from the previous study in this area (Adelana et al., 2014, Hydrological Processes, 29, 1630-1643). The effective porosity is unlikely to vary significantly and

given the nature of the aquifer materials is probably in the range from 0.03 to 0.2 (Morris and Johnson, 1967, U.S. Geological Survey Water-Supply Paper 1839-D, 42p). The uncertainties in the recharge estimates from TRR and WTF (if the specific yield is assumed to be close to the effective porosity) correspond to the uncertainties in the effective porosity. In terms of the TRR method, this is probably less than the uncertainty in h (the height of the upper part of the aquifer in which mixing occurs). In terms of the WTF method, the other issue around specific yield is more important. We can clarify the source of the porosity estimates in Section 3.3 and discuss the uncertainties more fully in Section 4.5.3.

Line 385 states $S_y$ is 'not well known' which is an understatement, as the parameter is highly uncertain. There is also a possibility of semi-confined conditions to develop at very shallow depths and that hydraulic loading could account for part of the water level response to rainfall.

We agree that $S_y$ is highly uncertain, as was discussed in the paper. Given that it is rarely measured and is not an invariant property, it is not surprising that there are errors in the WTF method. This is discussed extensively in the paper (sections 1.1.2 and 5).

If semi-confined conditions developed near the surface, one would expect rapid increases in the levels of WT following large rainfall events. We do not see the fast response to rainfall events, and the changes in WT levels are seasonal. We will make that clear in the revised paper.

6) CMB method is most reliant on the assumption of the long-term rate of Cl delivery and can only be applied in catchments with negligible runoff and sedimentary Cl inputs. How are the results sensitive to 8% runoff measurement from the catchments?

Like most semi-arid catchments, our study sites have minor surface water outflows. Not accounting for the export of Cl in surface water can lead to recharge rates being overestimated using the CMB method (we noted this at lines 357-360). This has a little overall effect on the conclusions as the recharge rates estimated from the CMB method are still lower than from the other two methods, and we can further emphasise that point.

Moreover, the stream discharge has probably increased due to the initial land clearing as is commonly the case throughout southeast Australia (Alison, 1990, Journal of Hydrology, 119, 1-20) and the streamflow prior to land clearing would have most probably been much lower. Additionally, the stream water is saline and most of the water and Cl is probably derived from shallow groundwater discharge - again this is generally the case in these types of streams. Because this is not direct surface runoff, it does not impact the recharge rate estimates. These latter two points are of general importance, and we can discuss them briefly.

7) The limitations of lumped parameter models (LPMs) should be discussed, as the dimensionless ratios assumed to vary over a very wide range (e.g. 0.05 to 1). Are the estimated residence times linearly related to these lumped parameters? Also, can it be clarified why the PEM and DM lumped parameter models were applied and not the exponential-piston flow model?

The lumped parameter models are used for two purposes, to estimate residence times using [14]C and to examine mixing using [3]H and [14]C. In terms of mixing, similar [3]H vs. [14]C trends are apparent using lumped parameter models and other models that predict the concentrations of the radioisotopes (e.g., the renewal rate calculations; Leduc et al., 2000, Earth and Planetary Science, 330, 355-361; Le Gal La Salle et al., 2001, Journal of Hydrology, 254, 145-156). For the mean residence times, the different lumped parameter models do yield different MRTs. However, this approach is much better than the much-used decay equation that ignores both mixing in the aquifer and variations in the

input function. The aims here were to broadly constrain the MRTs of the groundwater (i.e., to demonstrate that much of it is old). Other lumped parameter models (e.g., the EPM or the Gamma model) could have been used but yield MRTs that are similar to the ones that we report (lines 258-260). We will discuss the latter point briefly; however, it is not the main focus of the paper.

8) Clarify Line 295, regarding Cl/Br ratios 'and do not indicate that Cl is predominantly derived from rainfall and concentrated by evapotranspiration'.

Reviewer #2 also commented on this. This sentence was incorrect as written, it should have stated: "The observation that the Cl/Br ratios are significantly lower than those that would result from halite dissolution ($10^4$ to $10^5$: Kloppmann et al., 2001, Geochimica et Cosmochimica Acta, 65, 4087-4101; Cartwright et al., 2004, Applied Geochemistry, 19, 1233-1254; Cartwright et al., 2006, Chemical Geology, 231, 38-56) and do not increase with increasing salinity indicates that Cl is predominantly derived from rainfall". We will correct it in the revised paper.

9) Schematic cross-sections could help explain the relationship between regional vs. riparian groundwater. An additional map that shows the regional catchment context of the catchment divides for groundwater vs. surface water would also be helpful, as the current mapping provides very large scale and small-scale maps.

We will add cross-sections to better show the context and the hydrogeology. The catchments are at the top of a major surface water divide and the groundwater divide will correspond to the surface water divide.

10) Mean residence times, estimated from both $^3$H and $^{14}$C, were ~4K in pasture and ~24K in the forest. Yet the forest was planted the only ~20 years ago, after ~160 years of pasture. The CMB method suggests chloride accumulation over ~10K years of rainfall inputs, to account for relatively high salinities. These differential time scales should be discussed further.

This is an important point, and we will discuss it further. The time taken for the $^3$H activities to achieve steady-state is ~$1/R_N$ (i.e., the average residence time of the water in the well-mixed zone at the top of the aquifer). The initial land clearing should thus be evident in the TRR estimates, but the later revegetation may not be. Hence, in the cleared catchment, we would expect that WTF and TRR estimates agreed, but in the revegetated catchment, we may still be in the lag period where the $^3$H activities are showing transient behaviour between different recharge rates. This may also explain the observations in Comment 3. We will discuss this more explicitly.

---

## Short Comment (SC2) · 19 Jun 2020

We thank the reviewer for their helpful comments, our responses are below (in blue)

Overall, the paper is well written, and the results are solid. However, the most critical point I see is that the study does not provide a broader context. How do the results impact water management for the study area and for the region? Regarding the uncertainty in the estimated recharge rates and spatial and temporal variability, it is not obvious to me how sustainable water resource management can set up. Perhaps the authors have some thoughts about this problem and might provide some suggestions.

This was also raised by Reviewer #1. Our aim in this study was a better understanding of recharge rates in this area and assessing the uncertainties and limitations of commonly-applied recharge methods in general. As we explained, estimating recharge is important to understand the hydrogeology of semi-arid regions and this study has relevance to other regions where these techniques are used. Groundwater is not extensively used in this catchment, and our comments on sustainability were a more general recognition that groundwater can be a vital resource in semi-arid areas. In terms of the specific need to understand recharge in this catchment, defining whether recharge rates changed with land-use changes are important (especially the potential impacts on waterlogging and streamflow caused by changing water table elevations). We will clarify the aims and implications in the final paper.

Moreover, if one of the objectives of this study is to assess and compare uncertainty in the methods, then this has to be more elaborated and systematically compared. In addition, these results should be compared to similar studies.

We compared our results with other studies (Section 5, lines 439-441); e.g., Dean et al., 2015, Hydrology and Earth System Sciences, 19, 1107–1123; Perveen, 2016, http://hdl.handle.net/1959.9/560005); Cartwright et al., 2007, Journal of Hydrology, 332, 69-92; Crosbie et al., 2010, Hydrology and Earth System Sciences, 14, 2023-2038. These studies specifically mentioned that the WTF method overestimates recharge rates. We will emphasise this more in the discussion and introduction.

Furthermore, I miss a conceptual model which describe the processes. That can be a schematic figure or a cross-section describing the different flow systems and geochemical signatures.

We will add a cross-section to help understand the hydrogeology of the area and the processes.

Some further comments and suggestions are provided below.

Introduction: Personally, I believe that the study objectives should be clearly communicated in 1. Introduction. I found it a bit confusing to get information about the different methods before knowing the target of the study.

Although they normally appear at the end of the introduction, we can move the objectives to earlier in the introduction (before where we discuss the methods). This will help the reader understand how that discussion relates to the study areas. We will also make it clear how this study contributes to a general understanding of recharge in semi-arid areas.

Line 48ff. Not only in semi-arid areas recharge varies in space and time. Also in humid areas, recharge can be considerable spatially and temporally different (see, for example, Moeck et al., 2020 and Mohan et al., 2018, among many others) Line 50ff:

This is certainly the case. Although this paper is not a general review of recharge methods, we will note this in the introduction.

You could add Darcy methods, soil moisture methods, heat tracers, baseflow separation techniques, empirical relationships, etc. for completeness of the provided list (see for instance Healy, 2010, Walker et al., 2019).

**While a comprehensive review of recharge processes would be out of place in this paper, we will mention these other methods for completeness.**

Section 1.1.1. When residence times are around ~25000 years, how likely is that all Cl is originating from rainfall only and the impact of runoff can be neglected. This is more of a question rather than a critic. You already indicate based Cl/Br ratios that evapotranspiration rather than halite dissolution is the main process in controlling groundwater salinity but would be the error in estimated recharge rates if a small amount of Cl is not only originating from precipitation?

This method does assume that all the Cl is derived from rainfall. These are upland catchments with ephemeral stream systems, located at the top of a major regional catchment divide so the catchments do not receive any Cl input from sources other than precipitation. In the study area, the Cl/Br ratios, and the lack of halite in the soils and bedrock make this the case (regardless of the residence time of the waters), as discussed in Section 4.2. In general, it is also the case for other semi-arid areas in southeast Australia where recharge rates have been calculated (e.g., Cartwright et al., 2007, Journal of Hydrology, 332, 69-92), and similar recharge rates could be estimated using Br rather than Cl. However, it is something that needs to be tested whenever this method is used. We will discuss this assumption further in the introduction to the revised paper.

Line 85ff: In the study area with an actual ET of ~600 mm/a, to what depth can ET impact be observed. I am asking because I am not sure if the observation wells 3008 (depth 1.3, pasture) and 3657 (depth 2.5 m, forest with deeper root zones) can be reliably used by applying the water table fluctuation method, although I have to note that the estimated rates seem to be in the same range as for the other observation points.

The recharge rates using the WTF method were calculated for those bores because they show a clear seasonal variation (lines 222-223). We note that the effect of ET is likely small during winter when radiation and temperatures are lower and rainfall is larger. Additionally, the bores in the plantation are installed near the stream where trees are not planted to create a buffer zone and limit the effect of the plantation on streamflow. Thus, it is reasonable to consider the effect of evapotranspiration on the magnitude of the groundwater fluctuations to be low and possibly negligible. We will clarify this in the revised paper.

Line 295-297: Maybe I misunderstood something here, but did you not indicate that all Cl is delivered by rainfall (e.g. Line 351). Please check the statement and maybe reformulate the second part of the sentence.

Reviewer #1 also commented on this. This sentence was incorrect as written, it should have stated: "The observation that the Cl/Br ratios are significantly lower than those that would result from halite dissolution (104 to 105: Kloppmann et al., 2001; Cartwright et al., 2004; Cartwright et al., 2006, Chemical Geology, 231, 38-56) and do not increase with increasing salinity indicates that Cl is predominantly derived from rainfall". We will correct it in the revised paper.

Line333ff: Not clear. Please explain why it is not possible.

It is because it would require an initial a14C that is not possible. All the samples with measurable 3H that lie on the 3H vs. 14C covariance curves will be less than 200 years old. Over that time span, there has been negligible decay of 14C, and the initial a14C of the sample can be calculated by mass balance

(it is the measured  $a^{14}C/q$ ). Consider a sample with a measured  $a^{14}C$  of 95 pMC; if there were 10% contribution of DIC from  $^{14}C$ -free calcite dissolution (q = 0.9), it would imply an initial  $^{14}C$  activity of 106 pMC (which is plausible for water recharged during the bomb-pulse period). However, if we were to propose that q = 0.7, then the initial  $a^{14}C$  would need to be 136 pMC. This exceeds the highest  $a^{14}C$  recorded in soil CO2 of ~120 pMC (Jenkinson et al., 1992, Soil Biology and Biochemistry, 24, 295-308; Kuc et al., 2004, Geochronometria, 23, 45-50; Tipping et al., 2010, Geoderma, 155, 10-18) and so is implausible. The cited papers explain this; however, we will add a fuller explanation to the text.

Line392ff: Just from Fig.4 it is not possible to identify the samples. Perhaps you can better highlight these samples in Fig. 4 or provide the link to Table 1.

We can better highlight the samples that show the mixing in the figures, and also refer to Table 1 in the caption and text where this information is also shown.

Apart from that, I am wondering that location 3663 do not show mixing with older groundwater, even though it is one of the deepest locations (~25m) and based on the drawn picture with the stratification I was expecting that older groundwater exists.

While that may be expected, the observations from the 14C and 3H imply that little mixing has occurred at this locality. Although bore 3663 is deep, this is largely due to the topography, rather than depth below the water table (the screen is approx. 10-15 m below the water table). Location 3663 is in the regional recharge area and the groundwater in this area is expected to be relatively young. The groundwater flow here is likely to be downwards with little mixing with older laterally flowing groundwater.

Moreover, the decline in the groundwater level for 3663 is uniformly over the monitoring period, which is in contrast to the wells 3657 and 3669. But for all these the TRR is applied and no mixing is assumed. Could you please elaborate more on these differences?

Bore 3663 is the only one that is actually in the forest. The other two (and most others) are in cleared areas between the stands of trees. Since the water levels probably respond to recharge over areas of a few m2 (Scanlon et al., 2002, Hydrogeology Journal, 10, 18-39), the difference probably reflects the more limited recharge in the forest areas (e.g., lines 473-475). Some recharge will occur at all the sites since the aquifers are unconfined. We will explain the distribution of recharge better in the revised paper.

Line 491ff: Yes, I absolutely agree, and this is an important point. The question of which arises from the results is how we can set up sustainable water resource management than when we have such a spatial and temporal variability as well as uncertainty on ~500 ha. Perhaps the authors have some thoughts about this problem and might provide some suggestions.

As we replied to the other reviewer, our aims in this study was a better understanding of recharge rates in this area and semi-arid areas in general. This is important as we explained to understand the hydrogeology of semi-arid regions. Groundwater is not extensively used in this catchment, and our comments on sustainability were more general recognising that groundwater is a vital resource in semi-arid areas. In terms of the specific need to understand recharge in this catchment, the possibility that recharge rates change with land-use change is important (especially the potential impacts on waterlogging and streamflow). We will clarify both these matters in the final paper.

Line 495: Although I agree that physical-based models are useful tools, the models will likely not represent in every detail the recharge processes because of the lack of observations although a

relatively high density of data exists and information about the subsurface heterogeneity are missing (in both, x, y and z-direction). Thus, calibration is required which will also lead to uncertainty. For me, this part sounds like we always should use physical-based models, and then we get the right recharge rates which are obviously not true, even though the models are very powerful. Please reformulate.

We will modify this phrase not to give the impression that models are always correct and can be used to estimate correct recharge rates. This was not the intention of this paragraph, which rather meant to stress the uncertainties associated with the use of measured recharge rates as boundary conditions for groundwater models. Integrated models calculate recharge rates and can support experimental studies by providing an additional estimate of recharge rates.

Figure 1: If available, it would be useful to add the precipitation on top of the graphics (second y-axis).

This relates to Fig. 2. We will add the weekly rainfall to this figure.

Also, why does well 3658 show an increase?

Bore 3658 shows very little response over the monitoring period (head levels vary by <1.4 m). The bore monitors groundwater at ~16 m in the lower part of the catchment and probably does not record the short-term recharge. The actual water level measurements from 3658 are not very reliable because the formation has low permeability so two screens were installed in the completion. The upper screen was placed in what subsequently appears to be a perched zone that periodically dries up and the lower screen in a deeper zone.

---

## Referee Comment (RC3) · Anonymous Referee #3 · 15 Jul 2020

This is an interesting study comparing different methods of estimating recharge. It is well written and organised, but needs more thought in the interpretation.

1. There is some confusion in the paper about specific yield. Specific yield is, as the authors quote, "the volume of water that an unconfined aquifer releases from storage per unit surface area of aquifer per unit decline in the water table". This is the water that drains from the aquifer under the influence of gravity as the water table falls. As the water table falls, some water remains in the smaller pore spaces and as rims and menisci around grains. This definition does not "ignore the moisture in the unsaturated zone held in and above the capillary fringe". The moisture in the unsaturated zone does not drain significantly. The capillary fringe is saturated (not unsaturated) and moves downward with the watertable at the same rate. Therefore the reason that the

WTF method of estimating recharge gives unrealistically large values in this study is not the result of "the presence of moisture in the unsaturated zone and capillary fringe .. reduc(ing) the effective values of Sy".

2. There is also some confusion about porosity and effective porosity. Effective porosity is the porosity through which fluid can flow and is almost always less than total porosity. Effective porosity is generally similar to, or slightly less than, specific yield, as shown by comparison of Sy values from pumping tests and ne values from Darcy's Law (ne = Ki/v). In this study, values of Sy, n and ne are used interchangeably, and the authors need to correct this or at least explain why they used these values. Previous studies found Sy values of 0.03 to 0.1 and mean porosity of 0.1-0.15; in this paper values of Sy of 0.03 to 0.1 were used for the WTF method, whereas for the mass balance calculation (line 365) and the TRR method, n values of 0.03 to 0.1 were used. The latter are likely to be too low and will make the TRR numbers calculated also too low.

3. Note that "if the soil becomes fully saturated due to the rise of the capillary fringe", the top of the capillary fringe is at the ground surface and therefore no recharge can occur. Small recharge events cannot "produce significant and rapid increases in the head". This has no effect on the amount of water that can drain from the aquifer when the watertable drops, so Sy does not become "close to 0".

4. Another puzzling aspect is the definition of b for calculating TRR; b is "the thickness of the upper part of the aquifer system that receives annual recharge". Is this the part of the aquifer that is subject to water table fluctuations, i.e. is b equal to the maximum fluctuation? If this is the case, why not use this value? In this paper b is estimated from chemical stratification of regional groundwater (p. 18). But if the groundwater is stratified, then this could be because the upper part is not recharging the lower part, i.e. there are two separate aquifers. Alternatively, the chemical stratification could reflect the difference in recharge since clearing of native vegetation in the area. In either case, use of chemical stratification to estimate b is unjustified, and the presence of chemical stratification has implications for the CMB calculations; there should be

separate calculations for the upper and lower groundwater.

5. The forest bores show relatively small seasonal fluctuations compared to pasture bores, and some show no fluctuations at all (Fig 2). Yet the WTF recharge values for the forest are the same as those for the pasture (Fig 6). This seems very unlikely and requires explanation.

6. The WTF values calculated are not just unlikely and higher than expected, they are impossible. Recharge of this magnitude would imply that the vegetation was not extracting significant levels of water, and the consistent drop in the watertable beneath the forest shows that this is not the case.

7. The authors note that "there has been a rise in the water table caused by the increased recharge, and in some cases increased drainage in the streams"; what is the evidence for this in the study area? This topic has been much discussed in the Australian groundwater literature, and needs more discussion and explanation, with comparison with other areas in SE Australia.

8. Rainfall was sampled for tritium content. The sampling method needs to be briefly described and the results given in Table S1 (not a single average value).

9. The aquifer is described as "silty clay to coarse-grained sediments" and as comprising "interlayered clays and silts". Silts are not coarse-grained and the porosity values (o.1-0.15) suggest sandy sediments. The authors need to resolve this.

10. There are a few small grammatical/spelling errors: lines 107, 263, 295-296, 342, 358, 429.

11. Fig 2 would be better plotted as depth bgs.

---

## Author Comment (AC1) · 28 Jul 2020

*This is an interesting study comparing different methods of estimating recharge. It is well written and organised, but needs more thought in the interpretation.*

*1. There is some confusion in the paper about the specific yield. Specific yield is, as the authors quote, "the volume of water that an unconfined aquifer releases from storage per unit surface area of aquifer per unit decline in the water table". This is the water that drains from the aquifer under the influence of gravity as the water table falls. As the water table falls, some water remains in the smaller pore spaces and as rims and menisci around grains. This definition does not "ignore the moisture in the unsaturated zone held in and above the capillary fringe". The moisture in the unsaturated zone does not drain significantly. The capillary fringe is saturated (not unsaturated) and moves downward with the water table at the same rate. Therefore, the reason that the WTF method of estimating recharge gives unrealistically large values in this study is not the result of "the presence of moisture in the unsaturated zone and capillary fringe .. reduc(ing) the effective values of $S_y$".*

We agree with the point raised in this comment and it is what we tried to explain in Section 1.1.2. Many applications of the WTF method have assumed a value for $S_y$ that is close to the effective porosity. As the reviewer correctly points out, the value of $S_y$ needs to take into account the moisture content of the unsaturated zone and will be less than the effective porosity. This was discussed in detail by Crosbie et al. (2005, 2019) as noted in the text. The consequence of assuming that $S_y$ is equivalent to the effective porosity is to overestimate recharge using the WTF method (as discussed in Section 5). Nevertheless, many studies using the WTF method make this assumption, and this is something that we wanted to draw attention to. This section will be clarified to emphasise that what we were discussing a common assumption made around $S_y$ when using this method.

*2. There is also some confusion about porosity and effective porosity. Effective porosity is the porosity through which fluid can flow and is almost always less than total porosity. Effective porosity is generally similar to, or slightly less than, specific yield, as shown by the comparison of $S_y$ values from pumping tests and $n_e$ values from Darcy's Law ($n_e$ = Ki/v). In this study, values of $S_y$, n and $n_e$ are used interchangeably, and the authors need to correct this or at least explain why they used these values. Previous studies found $S_y$ values of 0.03 to 0.1 and mean porosity of 0.1-0.15; in this paper values of $S_y$ of 0.03 to 0.1 were used for the WTF method, whereas for the mass balance calculation (line 365) and the TRR method, n values of 0.03 to 0.1 were used. The latter is likely to be too low and will make the TRR numbers calculated also too low.*

Again we agree with the reviewer and will clarify this in the paper. As discussed above, the assumption that $S_y$ is similar to the effective porosity is one that is

commonly made for the WTF calculations but one that is probably incorrect (as we discussed in the paper). The values of porosity used in this study are those from Adelana et al. (2014) and are similar to typical values for these aquifer materials. We can discuss the uncertainties in the TRR recharge estimates arising from these values in Section 5; however, given the nature of the aquifer materials, they are unlikely to be significantly higher and the uncertainties arising from having to estimate b and the input function of $^3$H are probably greater. Overall, the TRR recharge rates are still considerably higher than those from the CMB method but lower than the WTF estimates.

*3. Note that "if the soil becomes fully saturated due to the rise of the capillary fringe", the top of the capillary fringe is at the ground surface and therefore no recharge can occur. Small recharge events cannot "produce significant and rapid increases in the head". This has no effect on the amount of water that can drain from the aquifer when the water table drops, so $S_y$ does not become "close to 0".*

This discussion was taken from Gillham (1984). The water levels that are measured in the bores correspond to the water table and not the top of the capillary fringe, which is not under positive pressure. As small amounts of water are added at the top of the capillary fringe, part of that water becomes pressurised and the head increases. As noted above, we probably over-complicated this discussion and the main point (which is that many applications of the WTF method misassign the $S_y$ value) got lost. We will omit this detail as it distracts from that point.

*4. Another puzzling aspect is the definition of b for calculating TRR; b is "the thickness of the upper part of the aquifer system that receives annual recharge". Is this the part of the aquifer that is subject to water table fluctuations, i.e. is b equal to the maximum fluctuation? If this is the case, why not use this value? In this paper, b is estimated from chemical stratification of regional groundwater (p. 18). But if the groundwater is stratified, then this could be because the upper part is not recharging the lower part, i.e. there are two separate aquifers. Alternatively, the chemical stratification could reflect the difference in recharge since the clearing of native vegetation in the area. In either case, the use of chemical stratification to estimate b is unjustified, and the presence of chemical stratification has implications for the CMB calculations; there should be separate calculations for the upper and lower groundwater.*

We defined b in section 1.1.3 and 4.5.3 using observations of chemical stratification. The value of 1 to 5 m is the distance at the top of the aquifer over which the groundwater chemistry is relatively uniform. These values of b are similar to those proposed elsewhere (Le Gal La Salle et al., 2001; Cartwright et al., 2007). We discuss the uncertainties in the b values in section 4.5.3; increasing the depth of the

zone of active mixing at the top of the aquifer to 10 m (which is unlikely given the observations of the depths over which the geochemistry varies) would increase the recharge rates but the differences between CMB, WTF and TRR recharge rates remain significant. Using the WTF to estimate b at the top of the aquifer is not generally done; however, it would yield a similar estimate of b, and we can discuss this.

There is no evidence of two separate aquifers; the bore logs do not indicate any major low K layers and the groundwater generally contains measurable $^3$H at depths of up to 29 m (Table S1). This precludes the presence of a deeper groundwater system that is isolated from the shallower part of the aquifers. The joint use of $^3$H and $^{14}$C activities allows macroscopic mixing of old and recently recharged groundwater in the aquifers to be tested (Fig. 4), which also implies a single flow system and adds confidence to the TRR calculations.

In terms of the CMB technique, if recent recharge rates have changed due to land clearing, the Cl concentrations in the upper part of the aquifer may be lower (this might be expected mainly in the pasture catchment). However, the Cl concentrations of the shallow and deeper groundwater overlap and there is no correlation between Cl and $^3$H (see below). This probably reflects the timescale of Cl delivery. Because Cl in saline groundwater may take several thousand years to accumulate, recent changes in evapotranspiration (which reduce the Cl concentrations) are not yet visible in the groundwater. Thus, the CMB recharge rates represent the average of those over the last several hundred to thousands of years. This has also been noted elsewhere in SE Australia (Cartwright et al., 2007; Dean et al., 2015). It is an interesting point that we can discuss in the revised manuscript.

[Figure]

[Figure]

*5. The forest bores show relatively small seasonal fluctuations compared to pasture bores, and some show no fluctuations at all (Fig 2). Yet the WTF recharge values for the forest are the same as those for the pasture (Fig 6). This seems very unlikely and requires explanation.*

In the Forest, the annual variations of the head in bores 3656, 3657 & 3669 are up to 3 m, which is similar to many of those in the Pasture catchment (Fig. 2). These are the bores that yield high WTF recharge rates. As we discussed in section 5.1, the trees cover ~62% of the forest catchment, and many of the bores are in cleared areas between the stands of trees (Fig. 1a). So, the recharge rates may not be representative of the forest as a whole and are similar to the cleared areas in the pasture. This is discussed in Section 5.1, but we will make sure that this is clear in the revised paper.

*6. The WTF values calculated are not just unlikely and higher than expected, they are impossible. Recharge of this magnitude would imply that the vegetation was not extracting significant levels of water, and the consistent drop in the water table beneath the forest shows that this is not the case.*

This is what we concluded in the paper (Section 5). As we discussed above, the WTF method commonly uses an estimate of $S_y$ that is based on the effective porosity. As noted by this reviewer and as we noted in the Introduction, this is one of the failings of the method. The consequence of doing this is to overestimate $S_y$ and to thus overestimate the recharge rates using the WTF method. Other studies have demonstrated that the WTF method yields higher recharge rates (Cartwright et al., 2007; Crosbie et al., 2010) and Crosbie et al. (2019) discussed this topic more generally. Nevertheless, the WTF method with this assumption of $S_y$ remains widely

used as we also discussed. We also explored other reasons why the WTF method yields high recharge rates (e.g., focussed recharge and the subsequent evaporation of water from the water table - Section 5). While these are also issues, the incorrect assumptions around the $S_y$ are probably more serious.

*7. The authors note that "there has been a rise in the water table caused by the increased recharge, and in some cases increased drainage in the streams"; what is the evidence for this in the study area? This topic has been much discussed in the Australian groundwater literature, and needs more discussion and explanation, with the comparison with other areas in SE Australia.*

The Gatum area is one of many in SE Australia that was identified as being impacted by dryland salinity due to land clearing and rising water tables (Clark and Harvey, 2008: Dryland salinity in Victoria in 2007, Department of Primary Industries Report). The area has common saline discharge to streams and local salt scalds. The bore monitoring and streamflow network were set up in the pasture catchment in this area on that basis. During the Millennium Drought in the first decade of the century, the water table levels dropped considerably and the emphasis on soil salinity diminished. The focus of water management in the area switched from salinity to water availability and the effect of land use on the water balance of these catchments. Accordingly, monitoring in the forest was set up to assess the subsequent impact of the tree plantation on the groundwater and surface water. We can add these details to the study area and emphasis that it is typical of many of these areas in SE Australia.

*8. Rainfall was sampled for tritium content. The sampling method needs to be briefly described and the results given in Table S1 (not a single average value).*

A one-year aggregated rainwater sample was collected in a narrow-mouthed container with an open funnel. We periodically removed the sample from the container and stored it in the lab (adding the subsequent rainfall to that sample). The value in Table S1 is the aggregate value, not an average. We will clarify this in the revised manuscript.

*9. The aquifer is described as "silty clay to coarse-grained sediments" and as comprising "inter-layered clays and silts". Silts are not coarse-grained and the porosity values (0.1-0.15) suggest sandy sediments. The authors need to resolve this.*

The aquifer materials are mainly silts to coarse-grained weathered ignimbrites with minor discontinuous clay layers. These are described by Brouwer & Fitzpatrick

(2002) and Adelena et al. (2014) who also report aquifer properties such as porosity. We will be consistent in the revised paper.

*10. There are a few small grammatical/spelling errors: lines 107, 263, 295-296, 342, 358, 429.*

We will correct grammatical/spelling errors in these lines.

*11. Fig 2 would be better plotted as depth bgs.*

Because we use Fig 2 to discuss heads in the catchment and this figure links to the head values in Fig. 1b, we prefer to leave this as it is. The individual bore hydrographs are also more easily seen on this version of the Figure. We will add rainfall to Fig. 2 as requested by the other reviewers.

---

## Author Comment (AC2) · 28 Jul 2020

Our response to referee 1 comments are below.

---

## Author Comment (AC3) · 28 Jul 2020

Our response to referee 2 comments are below.

———————————————

[Figure]

---

## Author Response (AR1)

We thank the reviewers for their helpful comments. As requested by the Associate Editor, we have made the conclusions to the paper more specific and discussed how our study here is important for a general understanding of processes (e.g. lines 818-850). Our responses are below (in blue), and line numbers refer to the track-changes version of the paper. We also reorganised various sections of the paper by

- Moving the objectives earlier in the Introduction
- Moving the sections on mean residence times and recharge rates into the Discussion (as these now include more discussion that interprets the results)
- Moving the more general aspects of the discussion to the Conclusions

**Reviewer 1**

How does this study inform the sustainable management of groundwater (from the opening line of the abstract)? The description of the study area does not mention groundwater use lower in the catchment, or reference to estimate of sustainable yields on a larger scale. A context of groundwater management issues in the region is not provided.

The study took place in two catchments instrumented to study the effects of land-use changes on groundwater and surface water systems and resources. The aims of our study were a better understanding of the uncertainties and limitations of commonly-applied recharge methods using these well-instrumented catchments as examples. As we explained in the Introduction (lines 75-92), estimating recharge is important to understanding the hydrogeology of semi-arid regions and this study has relevance to other regions where these techniques are used. In particular, as discussed below, the WTF method remains widely used but is prone to large uncertainties.

Groundwater is not extensively used in the catchments studied, and our comments on sustainability were a more general recognition that groundwater can be a vital resource in semi-arid areas. We have removed these points from the Abstract, Introduction and Conclusions.

In terms of the specific need to understand recharge in these catchments, defining whether recharge rates changed with land-use changes is important (especially the potential impacts on waterlogging and streamflow caused by changing water table elevations). We clarified these aims in the Introduction (lines 121-129) and have better reflected the outcomes of this study in the Conclusions (lines 818-830).

How do the authors reconcile their view of the importance of recharge estimates with the 'water budget myth'? A related myth that sustainable development of groundwater resources can be defined by groundwater residence times has recently been highlighted by Ferguson et al. 2020, citing classic papers on the water budget myth.

This is beyond what we can easily discuss in this paper. We agree that defining sustainable yields is difficult (and these may be a flawed concept); however, understanding recharge is a critical part of assessing the impacts of groundwater use (e.g., Gleeson et al., 2012, Nature,

11295). We have removed the references to sustainability from the introduction and conclusions as it is not something that is specifically discussed.

The use of "residence times" also differs between this paper and Ferguson et al. (2020). We use it to refer to individual samples rather than the average age of water in the aquifer (sometimes called the turnover time). We avoid using the term "age" for specific samples as it is not a valid concept for most groundwater and gives a misleading impression (Suckow et al., 2014, Applied Geochemistry, 50, 222–230).

Specific suggestions are provided below.

1) The objectives of the study were to examine uncertainties in varying methods of estimating recharge. However, there is no discussion of how the method comparison is similar or distinct from other recharge studies in semi-arid areas. Have other studies also found the WTF method overestimates recharge for example?

This was mentioned in the original version of the paper (Section 5) but we have moved this to the Conclusions (lines 797-830) to give it more prominence. We have made reference to previous studies (Cartwright et al., 2007, Journal of Hydrology, 332, 69-92; Crosbie et al., 2010, Hydrology and Earth System Sciences, 14, 2023-2038; Dean et al., 2015, Hydrology and Earth System Sciences, 19, 1107–1123; Perveen, 2016, http://hdl.handle.net/1959.9/560005) that show that the WTF method generally overestimates recharge rates, mainly due to specific yields being overestimated (for reasons discussed in Section 5.2.1).

2) Comparing methods for recharge rates is interesting, but the authors argue (Line 481 in the previous version) that it is 'fundamentally important to assess the impacts of land clearing'. Why?

Understanding the recharge changes is important for understanding the rise of the water table and consequent impacts of salinization of soils and rivers. We have indicated that in the Introduction (lines 80-90) and discussed it more explicitly in the Discussion (Section 5.3, lines 740-797) and the Conclusions (lines 840-850).

Inevitably, understanding the impacts of land clearing necessitates using different methods to understand pre- and post-land clearing recharge. Pre-land clearing recharge is commonly estimated using the CMB or longer-lived radioisotopes (e.g., $^{14}$C). Modern recharge may be estimated using the WTF method or $^{3}$H. However, this assumes that the rates from those different methods are all broadly correct (or at least comparable), which is probably not the case (as we discussed in the Conclusions).

3) Section 5.1 (in the previous version) on the impacts of reforestation only considers the TRR method, which surprisingly does not find a significant difference in recharge between pasture and forest. Other evidence indicates the forest is using more water, so the study appears to demonstrate the limitations of recharge estimation methods?

The area was partially replanted in the past ~20 years. We discussed that the WTF method overestimates recharge rates and CMB method yields long-term recharge rates. $^{3}$H activities and the TRR method should be applicable to understanding the initial land-use changes, and

the recharge rates from this technique is higher (as is discussed in Section 5.3). There are two reasons why the TRR method may not detect changed to recharge rates following reforestation. Firstly, recharge estimates based on groundwater geochemistry are probably averaged over areas of a few 10's to 100's m$^2$ (Scanlon et al., 2002, Hydrogeology Journal, 10, 18-39). Commonly, the bores in plantation forests are in cleared areas that are larger than this, meaning that the recharge rates may not be representative of the whole plantation (we discussed this on lines 824-830). Secondly, the TRR method averages recharge rates over several years to decades, and so it may not yet detect the latest reforestation. We discussed this in Section 5.3 (lines 790-796).

4) How do the authors recommend these results inform groundwater modelling? Line 495 (in the previous version).

Popular groundwater models, such as MODFLOW, use recharge rates as a boundary condition at the water table. Isotope methods give estimates of recharge rates across large areas over a relatively long period of times dating back hundreds of years. Because the WTF method estimates recharge rates at smaller spatial scales for the years when data are available, it is often considered to be more appropriate to constrain the models. However, the large discrepancies in the values of recharge rates from different techniques and the overestimated recharge rates with the WTF method pose questions on the quantification of boundary conditions for groundwater models. We have noted that care must be used in assigning recharge rates as boundary conditions numerical models (lines 844-850).

As also suggested in the Conclusions (lines 831-838), the use of integrated surface and subsurface hydrogeologic models might overcome this issue and provide an additional tool for the estimation of recharge rates to support experimental analyses.

5) Both WTF and TRR rely on estimating the effective porosity (or effective specific yield). Mean porosity was previously reported as 0.15 and 0.1 respectively for the pasture and forested catchments but is unclear how this was determined, and how sensitive the WTF and TRR methods are to the range of possible values.

The values of porosity were taken from the previous study in this area (Adelana et al., 2014, Hydrological Processes, 29, 1630-1643). The effective porosity from 0.03 to 0.1 seems reasonable given the nature of the aquifer materials (e.g. Morris and Johnson, 1967, U.S. Geological Survey Water-Supply Paper 1839-D, 42p). The uncertainties in the recharge estimates from TRR and WTF using the values of effective porosity and effective specific yield are discussed explicitly in Sections 5.2.2 and 5.2.3.

Line 385 (in the previous version) states $S_y$ is 'not well known' which is an understatement, as the parameter is highly uncertain. There is also a possibility of semi-confined conditions to develop at very shallow depths, and that hydraulic loading could account for part of the water level response to rainfall.

We agree that specific yield is highly uncertain, as was discussed in the paper. Given that it is rarely measured and is not an invariant property, it is not surprising that there are errors in the WTF method. This is discussed extensively in the paper (Sections 1.2, lines 232-239, and

5.2.2, lines 687-693). As discussed in response to the other reviewers, we have reduced the discussion of specific yield in the Introduction (Section 1.2, lines 232-239). This material is well covered by previous studies (e.g., Gillham, 1984; Sophocleous, 1985; Healy and Cook, 2002; Crosbie et al., 2019); however, it is still the case that most recharge studies that use the WTF method make the oversimplifying assumption that the specific yield is spatially and temporally uniform (which is the point that we were trying to get across)

If semi-confined conditions developed near the surface, one would expect rapid increases in the levels of WT following large rainfall events. We do not see the fast response to rainfall events, and the changes in levels of WT are seasonal. We discussed this in Section 5.2.2 (lines 675-677).

6) CMB method is most reliant on the assumption of the long-term rate of Cl delivery and can only be applied in catchments with negligible runoff and sedimentary Cl inputs. How are the results sensitive to 8% runoff measurement from the catchments?

Like most semi-arid catchments, our study sites have minor surface water outflows. Not accounting for the export of Cl in surface water can lead to recharge rates being overestimated using the CMB method (discussed in Section 5.2.1, lines 619-625). This has a little overall effect on the conclusions as the recharge rates estimated from the CMB method are still lower than from the other two methods.

Moreover, the stream discharge has probably increased due to the initial land clearing as is commonly the case throughout southeast Australia (Alison, 1990, Journal of Hydrology, 119, 1-20) and the streamflow prior to land clearing would have most probably been much lower. Additionally, the stream water is saline, and much of the Cl exported by the streams is probably derived from shallow groundwater discharge; not from direct surface runoff. That component of Cl does not need to be corrected for in the CMB calculations. We discussed these issues in Section 5.2.1 (lines 619-623).

7) The limitations of lumped parameter models (LPMs) should be discussed, as the dimensionless ratios assumed to vary over a very wide range (e.g. 0.05 to 1). Are the estimated residence times linearly related to these lumped parameters? Also, can it be clarified why the PEM and DM lumped parameter models were applied and not the exponential-piston flow model?

We discussed the limitations briefly in Section 5.1 (lines 577-587). The lumped parameter models are used for two purposes: to estimate residence times using $^{14}$C and to examine mixing using $^{3}$H and $^{14}$C. In terms of mixing, similar $^{3}$H vs. $^{14}$C trends are apparent using lumped parameter models and other models that predict the concentrations of the radioisotopes (e.g., the renewal rate calculations; Leduc et al., 2000, Earth and Planetary Science, 330, 355-361; Le Gal La Salle et al., 2001, Journal of Hydrology, 254, 145-156). The aims were here to broadly constrain the MRTs of the groundwater (i.e., to demonstrate that much of it is old). Other lumped parameter models (e.g., the EPM or the Gamma model) could have been used but yield MRTs that are similar to the ones that we reported (Section 3.4, lines 457-459). As we noted on lines 578-580 use of lumped parameter models are considerably preferable to

using the decay equation that assumes piston flow and ignores variations in the $^{14}$C of the atmosphere.

8) Clarify Line 295 (now lines 509-511), regarding Cl/Br ratios 'and do not indicate that Cl is predominantly derived from rainfall and concentrated by evapotranspiration'.

We corrected the sentence (lines 507-511).

9) Schematic cross-sections could help explain the relationship between regional vs. riparian groundwater. An additional map that shows the regional catchment context of the catchment divides for groundwater vs. surface water would also be helpful, as the current mapping provides very large scale and small-scale maps.

We added cross-sections to better show the context and the hydrogeology (Figure 2 in the revised manuscript). The catchments are at the top of a major surface water divide, and the groundwater divide will likely correspond to the surface water divide.

10) Mean residence times, estimated from both $^3$H and $^{14}$C, were ~4K in pasture and ~24K in the forest. Yet the forest was planted the only ~20 years ago, after ~160 years of pasture. The CMB method suggests chloride accumulation over ~10K years of rainfall inputs, to account for relatively high salinities. These differential time scales should be discussed further.

This is an important point that we now discussed in Section 5.3 (lines 790-796). The time taken for the $^3$H activities to achieve steady-state is ~$1/R_N$ (i.e., the average residence time of the water in the well-mixed zone at the top of the aquifer). The initial land clearing should be evident in the TRR estimates, but the later revegetation may not be. Hence, in the cleared catchment, we would expect that WTF and TRR estimates agreed, but in the revegetated catchment, we may still be in the lag period where the $^3$H activities are showing transient behaviour between different recharge rates. This may also explain the observations in Comment 3.

**Reviewer 2:**

How do the results impact water management for the study area and for the region? Regarding the uncertainty in the estimated recharge rates and spatial and temporal variability, it is not obvious to me how sustainable water resource management can set up. Perhaps the authors have some thoughts about this problem and might provide some suggestions.

Our aim in this study is a better understanding of recharge rates in this area and assessing the uncertainties and limitations of commonly-applied recharge methods in general. As we have now made clear in the Introduction (lines 75-93) and the Conclusions (lines 839-846), estimating recharge is important to understand the impact of land-use change in semi-arid regions (especially the potential impacts on waterlogging and streamflow caused by changing water table elevations). As such, this study has relevance to other semi-arid regions.

Groundwater is not extensively used in the catchments studied, and our comments on sustainability were a more general recognition that groundwater can be a vital resource in semi-arid areas. As noted above, in response to Reviewer #1, we have removed that

discussion. In terms of the specific need to understand recharge, defining whether recharge rates changed with land-use changes is important for a complete understanding of the catchment water balance, and thus, land and water management.

If one of the objectives of this study is to assess and compare uncertainty in the methods, then this has to be more elaborated and systematically compared. In addition, these results should be compared to similar studies.

This was in the paper, but we have moved this discussion to the Conclusions to make it more prominent (lines 805-823). We compare our results with other studies in semi-arid areas (e.g., Dean et al., 2015, Hydrology and Earth System Sciences, 19, 1107–1123; Perveen, 2016, http://hdl.handle.net/1959.9/560005); Cartwright et al., 2007, Journal of Hydrology, 332, 69-92; Crosbie et al., 2010, Hydrology and Earth System Sciences, 14, 2023-2038). These studies also show that the WTF method yields higher recharge rates.

I miss a conceptual model which describe the processes. That can be a schematic figure or a cross-section describing the different flow systems and geochemical signatures.

We added cross-sections (Figure 2 of the revised manuscript), which will help understand the hydrogeology of the area and the processes.

Some further comments and suggestions are provided below.

Introduction: Personally, I believe that the study objectives should be clearly communicated in 1. Introduction. I found it a bit confusing to get information about the different methods before knowing the target of the study.

Although they normally appear at the end of the Introduction, we have moved the objectives to earlier in the Introduction (lines 121-129) before where we discussed the different methods to estimate recharge rates.

Line 48. Not only in semi-arid areas recharge varies in space and time. Also in humid areas, recharge can be considerable spatially and temporally different (see, for example, Moeck et al., 2020 and Mohan et al., 2018, among many others)

This is certainly the case. We have noted that estimating recharge rates, in general, is difficult (lines 103-105) and referred to the Moeck et al. (2020) study here and elsewhere.

Line 50: You could add Darcy methods, soil moisture methods, heat tracers, baseflow separation techniques, empirical relationships, etc. for completeness of the provided list (see for instance Healy, 2010, Walker et al., 2019).

We mentioned these other methods for completeness (lines 106-114).

Section 1.1.1: When residence times are around ~25000 years, how likely is that all Cl is originating from rainfall only and the impact of runoff can be neglected. This is more of a question rather than a critic. You already indicate based Cl/Br ratios that evapotranspiration rather than halite dissolution is the main process in controlling groundwater salinity but would be the error in estimated recharge rates if a small amount of Cl is not only originating from precipitation?

This method does assume that all the Cl is derived from rainfall. These are upland catchments with intermittent stream systems, located at the top of a major regional catchment divide; therefore, the catchments do not receive any Cl input from sources other than rainfall. In the study area, the Cl/Br ratios, and the lack of halite in the soils and bedrock make this the case (regardless of the residence time of the waters), as discussed in Section 4.2 (lines 509-514). In general, it is also the case for other semi-arid areas in southeast Australia where recharge rates have been calculated (e.g., Cartwright et al., 2007, Journal of Hydrology, 332, 69-92), and similar recharge rates could be estimated using Br rather than Cl. However, it is something that needs to be tested whenever this method is used.

Line 85: In the study area with an actual ET of ~600 mm/a, to what depth can ET impact be observed. I am asking because I am not sure if the observation wells 3008 (depth 1.3, pasture) and 3657 (depth 2.5 m, forest with deeper root zones) can be reliably used by applying the water table fluctuation method, although I have to note that the estimated rates seem to be in the same range as for the other observation points.

The recharge rates using the WTF method were calculated for those bores because they show a clear seasonal variation (Section 3.3, lines 407-411). The effect of ET on water tables is likely small during winter when radiation and temperatures are lower, and rainfall is larger (as we discussed on lines 684-686). Additionally, the bores in the plantation are installed in cleared areas near the stream where trees are not planted to create a buffer zone and limit the effect of the plantation on streamflow, which may also limit the impact of evapotranspiration.

Line 295-297: Maybe I misunderstood something here, but did you not indicate that all Cl is delivered by rainfall (e.g. Line 351; now Lines 618-619). Please check the statement and maybe reformulate the second part of the sentence.

We corrected the sentence (lines 505-507)

Line 333: Not clear. Please explain why it is not possible.

It is because it would require an initial $a^{14}C$ that is not possible. All the samples with measurable $^3H$ that lie on the $^3H$ vs. $^{14}C$ covariance curves will be less than 200 years old. Over that time span, there has been negligible decay of $^{14}C$, and the initial $a^{14}C$ of the sample can be calculated by mass balance (it is the measured $a^{14}C/q$). Consider a sample with a measured $a^{14}C$ of 95 pMC; if there were 10% contribution of DIC from $^{14}C$-free calcite dissolution (q = 0.9), it would imply an initial $^{14}C$ activity of 106 pMC (which is plausible for water recharged during the bomb-pulse period). However, if we were to propose that q = 0.7, then the initial $a^{14}C$ would need to be 136 pMC. This exceeds the highest $a^{14}C$ recorded in soil $CO_2$ of ~120 pMC (Jenkinson et al., 1992, Soil Biology and Biochemistry, 24, 295-308; Kuc et al., 2004, Geochronometria, 23, 45-50; Tipping et al., 2010, Geoderma, 155, 10-18) and so is implausible. We have explained this in Section 5.1 (lines 571-576).

Line 392: Just from Fig. 4 it is not possible to identify the samples. Perhaps you can better highlight these samples in Fig. 4 or provide the link to Table 1.

We highlighted the samples that show the mixing in Figure 5 (previously Fig. 4) and also referred to Table 1 in the and text (lines 695-697) where this information is also shown.

Apart from that, I am wondering that location 3663 do not show mixing with older groundwater, even though it is one of the deepest locations (~25m) and based on the drawn picture with the stratification I was expecting that older groundwater exists.

While that may be expected, the observations from the $^{14}$C and $^{3}$H imply that little mixing has occurred at this locality. Although bore 3663 is deep, this is largely due to the topography, rather than depth below the water table (the screen is approx. 10-15 m below the water table). Location 3663 is in the regional recharge area, and the groundwater in this area is expected to be relatively young. The groundwater flow here is likely to be downwards with little mixing with older laterally flowing groundwater. The cross-section (Fig. 2b) helps to illustrate this.

The decline in the groundwater level for 3663 is uniformly over the monitoring period, which is in contrast to the wells 3657 and 3669. But for all these, the TRR is applied and no mixing is assumed. Could you please elaborate more on these differences?

Bore 3663 is the only one that is actually in the forest. The other two (and most others) are in cleared areas between the stands of trees. Since the water levels probably respond to recharge over areas of a few m$^2$ (Scanlon et al., 2002, Hydrogeology Journal, 10, 18-39), the difference probably reflects the more limited recharge in the forest areas (this is now discussed in Section 5.3, lines 781-810). Some recharge will occur at all the sites since the aquifers are unconfined.

Line 491: Yes, I absolutely agree, and this is an important point. The question of which arises from the results is how we can set up sustainable water resource management than when we have such a spatial and temporal variability as well as uncertainty on ~500 ha. Perhaps the authors have some thoughts about this problem and might provide some suggestions.

As explained above the study is less concerned with sustainable water use than the impacts of a rising water table on salinization of soils and streams. We have discussed the implications more specifically on lines 788-830 and 839-980.

Line 495: Although I agree that physical-based models are useful tools, the models will likely not represent in every detail the recharge processes because of the lack of observations although a relatively high density of data exists and information about the subsurface heterogeneity are missing (in both, x, y and z-direction). Thus, calibration is required, which will also lead to uncertainty. For me, this part sounds like we always should use physical-based models, and then we get the right recharge rates which are obviously not true, even though the models are very powerful. Please reformulate.

We modified this discussion not to give the impression that models can be used to estimate recharge rates (lines 832-838). This was not the intention of this paragraph, which rather meant to stress the uncertainties associated with the use of measured recharge rates as boundary conditions for groundwater models. Integrated models with several simplifying assumptions and without accounting for the large spatial soil heterogeneity, can be used to support experimental studies by providing an additional estimate of recharge rates and suggest reasonable upper limits of recharge rates.

Figure 1: If available, it would be useful to add the precipitation on top of the graphics (second y-axis).

This relates to Figure 2. We added the rainfall to this figure (now Fig. 2a).

Also, why does well 3658 show an increase?

Bore 3658 shows very little response over the monitoring period (head levels vary by <1.4 m). The bore monitors groundwater at ~16 m in the lower part of the catchment and probably does not record the short-term recharge.

**Reviewer 3:**

1. There is some confusion in the paper about the specific yield. Specific yield is, as the authors quote, "the volume of water that an unconfined aquifer releases from storage per unit surface area of aquifer per unit decline in the water table". This is the water that drains from the aquifer under the influence of gravity as the water table falls. As the water table falls, some water remains in the smaller pore spaces and as rims and menisci around grains. This definition does not "ignore the moisture in the unsaturated zone held in and above the capillary fringe". The moisture in the unsaturated zone does not drain significantly. The capillary fringe is saturated (not unsaturated) and moves downward with the water table at the same rate. Therefore, the reason that the WTF method of estimating recharge gives unrealistically large values in this study is not the result of "the presence of moisture in the unsaturated zone and capillary fringe .. reduc(ing) the effective values of $S_Y$".

We agree with the points raised in this comment. As discussed in response to the other reviewers, we shortened our discussion of specific yield in the introduction (Section 1.2, lines 232-239). This material is well covered by previous studies (e.g., Gillham, 1984; Sophocleous, 1985; Healy and Cook, 2002; Crosbie et al., 2019) that we cite, and our longish discussion of that material was confusing. It remains the case though that most recharge studies that use the WTF method make the oversimplifying assumption that the specific yield is spatially and temporally uniform. Indeed, many assume that the specific yield is close to the effective porosity. We have clarified these points throughout the paper (Sections 1.2 and 5.2.2).

2. There is also some confusion about porosity and effective porosity. Effective porosity is the porosity through which fluid can flow and is almost always less than total porosity. Effective porosity is generally similar to, or slightly less than, specific yield, as shown by the comparison of $S_Y$ values from pumping tests and $n_e$ values from Darcy's Law ($n_e = Ki/v$). In this study, values of $S_Y$, n and $n_e$ are used interchangeably, and the authors need to correct this or at least explain why they used these values. Previous studies found $S_Y$ values of 0.03 to 0.1 and mean porosity of 0.1-0.15; in this paper values of $S_Y$ of 0.03 to 0.1 were used for the WTF method, whereas for the mass balance calculation (line 365 in the previous version) and the TRR method, n values of 0.03 to 0.1 were used. The latter is likely to be too low and will make the TRR numbers calculated also too low.

We agree with the reviewer and have clarified this in the paper (lines 232-238). As discussed above, the assumption that specific yield is similar to the effective porosity is one that is

commonly made for the WTF calculations but one that is probably incorrect (as is also discussed by Gillham, 1984; Sophocleous, 1985; Healy and Cook, 2002; Crosbie et al., 2019).

The values of porosity used in this study are those from Adelana et al. (2014) and are similar to typical values for these aquifer materials. We discussed the uncertainties in the TRR recharge estimates arising from these values in Section 5.2.3; however, the uncertainties arising from having to estimate b and the input function of $^3$H are greater (lines 716-740). Overall, the TRR recharge rates are still considerably higher than those from the CMB method but lower than the WTF estimates.

3. Note that "if the soil becomes fully saturated due to the rise of the capillary fringe", the top of the capillary fringe is at the ground surface and therefore no recharge can occur. Small recharge events cannot "produce significant and rapid increases in the head". This has no effect on the amount of water that can drain from the aquifer when the water table drops, so $S_y$ does not become "close to 0".

This discussion was taken from Gillham (1984). As we have noted above, we have simplified this in Section (1.2) and just noted that the common assumption made in the WTF method overestimates the specific yield and consequently recharge rates.

4. Another puzzling aspect is the definition of b for calculating TRR; b is "the thickness of the upper part of the aquifer system that receives annual recharge". Is this the part of the aquifer that is subject to water table fluctuations, i.e. is b equal to the maximum fluctuation? If this is the case, why not use this value? In this paper, b is estimated from chemical stratification of regional groundwater (p. 18: now p. 19-20). But if the groundwater is stratified, then this could be because the upper part is not recharging the lower part, i.e. there are two separate aquifers. Alternatively, the chemical stratification could reflect the difference in recharge since the clearing of native vegetation in the area. In either case, the use of chemical stratification to estimate b is unjustified, and the presence of chemical stratification has implications for the CMB calculations; there should be separate calculations for the upper and lower groundwater.

We defined b using observations of chemical stratification (lines 702-705). The value of 1 to 5 m is the distance at the top of the aquifer over which the groundwater chemistry is relatively uniform. These values of b are similar to those proposed elsewhere (Le Gal La Salle et al., 2001; Cartwright et al., 2007). We discussed the uncertainties in the b values in Sections 5.2.3 (lines 736-738) and 5.3 (lines 716-741); increasing the depth of the zone of active mixing at the top of the aquifer to 10 m (which is unlikely given the observations of the depths over which the geochemistry varies) would increase the recharge rates, but the significant differences between CMB, WTF and TRR recharge rates remain. This method of estimating b is the same as in the previous studies (including those of Leduc et al., 2000 and Le Gal La Salle et al., 2001) from which we derived the method.

There is no evidence of two discretely separate aquifers; the bore logs do not indicate any major low K layers, and the groundwater generally contains measurable $^3$H at depths of up to 29 m (Table S1). This precludes the presence of a deeper groundwater system that is not isolated from the shallower part of the aquifers. The joint use of $^3$H and $^{14}$C activities allows

macroscopic mixing of old and recently recharged groundwater in the aquifers to be tested (Fig. 5 and discussion on lines 554-559), and adds confidence to the TRR calculations. Many aquifers show some chemical stratification without consisting of discretely separated flow systems.

In terms of the CMB technique, if recent recharge rates have changed due to land clearing, the Cl concentrations in the upper part of the aquifer may be lower (this might be expected mainly in the pasture catchment). However, the Cl concentrations of the shallow and deeper groundwater overlap and there is no correlation between Cl and $^3$H (we have added Figures 4b & 4c to show this). This probably reflects the timescale of Cl delivery. Because Cl in saline groundwater may take several thousand years to accumulate, recent changes in evapotranspiration (which reduce the Cl concentrations) are not yet visible in the groundwater. Thus, the CMB recharge rates represent the average of those over the last several hundred to thousands of years. This has also been noted elsewhere in SE Australia (Crosbie et al., 2010; Cartwright et al., 2007; Dean et al., 2015). We have noted this in Section 5.3 (lines 748-751)

5. The forest bores show relatively small seasonal fluctuations compared to pasture bores, and some show no fluctuations at all (Fig 2: now Figs 3b, 3c). Yet the WTF recharge values for the forest are the same as those for the pasture (Fig 6: now Fig 7). This seems very unlikely and requires explanation.

In the forest, the annual variations of the head in bores 3656, 3657 & 3669 are up to 3 m, which is similar to many of those in the pasture catchment (Figs. 3b, 3c). These are the bores that yield high WTF recharge rates. As we discussed in Section 5.3 (lines 788-790), the trees cover ~62% of the forest catchment, and many of the bores are in cleared areas between the stands of trees (Fig. 1a). So, the recharge rates may not be representative of the forest as a whole and are similar to the cleared areas in the pasture.

6. The WTF values calculated are not just unlikely and higher than expected, they are impossible. Recharge of this magnitude would imply that the vegetation was not extracting significant levels of water, and the consistent drop in the water table beneath the forest shows that this is not the case.

This is what we concluded in the paper (Sections 5.2.2, lines 666-670 and the Conclusions, lines 804-807). The WTF method with the common assumption of a consistent specific yield remains widely used as we discussed. We also explored other reasons why the WTF method yields high recharge rates (e.g., focussed recharge and the subsequent evaporation of water from the water table – lines 681-685). While these are also issues, the inaccurate assumptions around the specific yield are probably more serious.

7. The authors note that "there has been a rise in the water table caused by the increased recharge, and in some cases increased drainage in the streams"; what is the evidence for this in the study area? This topic has been much discussed in the Australian groundwater literature, and needs more discussion and explanation, with the comparison with other areas in SE Australia.

The Gatum area is one of many in SE Australia that was identified as being impacted by dryland salinity due to land clearing and rising water tables (Clark and Harvey, 2008: Dryland salinity in Victoria in 2007, Department of Primary Industries Report). The area has common saline discharge to streams and local salt scalds. The bore monitoring and streamflow network were set up in the pasture catchment in this area on that basis. During the Millennium Drought in the first decade of the century, the water table levels dropped considerably and the emphasis on soil salinity diminished. The focus of water management in the area switched from salinity to water availability and the effect of land use on the water balance of these catchments. Accordingly, monitoring in the forest was set up to assess the subsequent impact of the tree plantation on the groundwater and surface water. We added these details to the study area (lines 327-335) and emphasised in the Conclusions (lines 840-980) that it is typical of many similar areas in SE Australia and elsewhere.

8. Rainfall was sampled for tritium content. The sampling method needs to be briefly described and the results given in Table S1 (not a single average value).

We briefly described this Section 3.1 (lines 371-373). The rainfall sample is an aggregate (i.e. successive samples were collected and mixed into a single sample), not an average.

9. The aquifer is described as "silty clay to coarse-grained sediments" and as comprising "inter-layered clays and silts". Silts are not coarse-grained and the porosity values (0.1-0.15) suggest sandy sediments. The authors need to resolve this.

The aquifer materials are mainly silt-sized to coarse-grained weathered ignimbrites with minor discontinuous clay layers. These are described by Brouwer & Fitzpatrick (2002) and Adelena et al. (2014) who also report aquifer properties such as porosity. We have clarified this in Section 2 (lines 296-314).

10. There are a few small grammatical/spelling errors: lines 107, 263, 295-296, 342, 358, 429 (now lines 241, 463-466, 509-511, 588, 625, 775).

We have corrected the grammatical and spelling errors on these lines.

11. Fig 2 (now Figs 3b, 3c) would be better plotted as depth bgs.

Because we use Figures 3b, 3c to discuss heads in the catchment and this figure links to the head values in Figure 1b, we prefer to leave this as it is. The individual bore hydrographs are also more easily seen on this version of the figure.

[revised manuscript text omitted]

---

## Author Response (AR2)

As the reviewer correctly states, the method of calculating b has been used by several previous studies that we cite in the paper (e.g., Leduc et al., 2000; Le Gal La Salle et al., 2001; Favreau et al., 2002; Cartwright et al., 2007). The conceptualisation of the model is that the upper part of the aquifer comprises a zone where a certain amount of water is added each year and the same amount leaks deeper into the groundwater system. The new water mixes with older water in the upper part of the aquifer and Eq. (3) is essentially a mixing equation. If land-use change occurs, then the proportion of water that is replaced each year would also change. Initially the system would not be in steady state but after a time it would adjust to the new renewal rate. As explained on lines 484-490, the residence time of water in the upper part of the aquifer is $\sim 1/R_N$ (where $R_N$ is the renewal rate, or the fraction of water replaced each year) and the system may take a similar amount of time to adjust. We only calculated recharge rates based on this method from the shallower groundwater (bores screened within a few metres of the water table) and used the $^{14}C$ vs $^3H$ activities to discount macroscopic mixing (which is also a requirement of using this technique). The deeper groundwater probably was recharged at a time when recharge rates were different; however, we do not use this method to calculate recharge rates from that groundwater.

If the geochemistry of groundwater in the upper layer of the aquifer were significantly and systematically different to that from the deeper layers, then the downward displacement of the geochemical interface could be used to estimate recharge rates. However, the differences (particularly the salinity) are not that systematic (i.e., the upper aquifer does not contain uniformly lower salinity water). This is probably due to irregularities in flow within the aquifers.

The suggestion to use the water level fluctuation to estimate b has not been done previously. However, it is logical as it may represent the zone that is receiving active recharge. We have added this to the paper (lines 449-453) and noted that it gives the same estimates of b.

While uncertainties in the value of b are significant, b cannot be so high that the renewal rate estimates will agree with those from the water table fluctuations (which is one of the main points of the comparison).

[revised manuscript text omitted]